# Black carbon-climate interactions regulate dust burdens over India revealed during COVID-19

Linyi Wei[1,7], Zheng Lu [2,7], Yong Wang [1✉], Xiaohong Liu [2✉], Weiyi Wang[3,4], Chenglai Wu [3], Xi Zhao [2], Stefan Rahimi[5], Wenwen Xia [1] & Yiquan Jiang[6]

India as a hotspot for air pollution has heavy black carbon (BC) and dust (DU) loadings. BC has been identified to significantly impact the Indian climate. However, whether BC-climate interactions regulate Indian DU during the premonsoon season is unclear. Here, using long-term Reanalysis data, we show that Indian DU is positively correlated to northern Indian BC while negatively correlated to southern Indian BC. We further identify the mechanism of BC-dust-climate interactions revealed during COVID-19. BC reduction in northern India due to lockdown decreases solar heating in the atmosphere and increases surface albedo of the Tibetan Plateau (TP), inducing a descending atmospheric motion. Colder air from the TP together with warmer southern Indian air heated by biomass burning BC results in easterly wind anomalies, which reduces dust transport from the Middle East and Sahara and local dust emissions. The premonsoon aerosol-climate interactions delay the outbreak of the subsequent Indian summer monsoon.

[1] Department of Earth System Science, Ministry of Education Key Laboratory for Earth System Modeling, Institute for Global Change Studies, Tsinghua University, Beijing 100084, China. [2] Department of Atmospheric Sciences, Texas A&M University, College Station, TX 77843, USA. [3] International Center for Climate and Environment Sciences, Institute of Atmospheric Physics, Chinese Academy of Sciences, Beijing 100029, China. [4] University of Chinese Academy of Sciences, Beijing 100049, China. [5] Institute of the Environment and Sustainability, University of California Los Angeles, Los Angeles, CA 90095, USA. [6] CMA-NJU Joint Laboratory for Climate Prediction Studies, Institute for Climate and Global Change Research, School of Atmospheric Sciences, Nanjing University, Nanjing 210023, China. [7] These authors contributed equally: Linyi Wei and Zheng Lu. ✉email: yongw@mail.tsinghua.edu.cn; xiaohong.liu@tamu.edu

Atmospheric aerosols including absorbing particles, e.g., black carbon (BC) and dust (DU), can scatter and absorb incoming solar radiation, thus heating the atmosphere and cooling the surface[1,2]. A changed radiative balance may further perturb the atmospheric general circulation[3–5]. South Asia, especially India, is a heavily-polluted region susceptible to anthropogenic aerosols such as BC produced locally from incomplete combustion of carbonaceous fuels (e.g., fossil fuels and biofuels)[2,6], biomass burning aerosols such as BC from crop residue burning (CRB)[7,8], and natural DU emitted locally from the Thar Desert and transported from remote deserts (e.g., the Middle East and Sahara)[9–12]. The accumulation of DU and BC in northern India (Indo–Gangetic Plain, IGP), acting as a heat pump, can reinforce rainfall during the premonsoon and summer monsoon seasons[13,14]. Also, BC and DU deposition on snow over the Tibetan Plateau (TP) and the Himalayan region can accelerate snow melting and decrease surface albedo[15–18]. However, detecting the BC and DU effects on the Indian climate remains challenging due to climate feedback and natural climate variability[19,20].

As the coronavirus disease 2019 (COVID-19) has been spreading worldwide, strict social, travel, and working restrictions (i.e., lockdown) have been launched in many nations to control the rapid spread of the disease. This rapid and dramatic air pollution reduction, especially those from the transportation and manufacturing sectors, provides a unique testbed for probing the effects of aerosols on climate[21]. Many studies have investigated the impacts of emission reductions on regional air quality and atmospheric composition, mostly focusing on China in January and February 2020[22,23]. India, as one epicenter of the pandemic, experienced the first nationwide lockdown for 14 h on March 22, 2020 followed by a 21-day lockdown starting from March 24, 2020[24]. Associated with the reduction of anthropogenic emissions in India in April-May[25–27], the observed $PM_{2.5}$ (particulate matters with diameters less than 2.5 μm) in northern India plunged to unprecedented low levels[24]. As a result, the Himalayas were visible from northern India in April 2020 for the first time in several decades[28]. Meanwhile, April and May each year are the peak season of winter crop harvesting (e.g., wheat and gram) and planting of vegetables in India[29]. Frequent CRB is mainly found in northwestern and southern India[30]. It has been reported that the CRB in April-May 2020 over northwestern and southern India has experienced sharp decreases and increases, respectively compared to the past two years[31]. This is because migrant laborers, who worked to transplant rice paddies in large-scale collective farms over northwestern India, returned to their individual farms over southern India due to the COVID-19 lockdown[32]. As evident from the Global Fire Emissions Database version 4 (GFED4.1s), CRB emissions in southern India are increased compared to 2015–2019 by as much as 0.06 mg/m²/day, while those in northwestern India decline with the comparable magnitude (Supplementary Fig. 1a). Surprisingly, besides changes of anthropogenic and biomass burning aerosols, DU in India experienced a record low in April-May 2020 based on the Cloud-Aerosol Lidar and Infrared Pathfinder Satellite Observations (CALIPSO) (Fig. 1a, e).

Since DU is a major natural aerosol species whose emissions are highly susceptible to surface winds but not directly affected by human activities, why did it decrease strongly in the premonsoon season (April-May) in 2020? Is it an indication of climate change-induced by anthropogenic and biomass burning aerosols—especially BC—over India? If so, how is the regional climate changed by the anomalies of BC in April-May 2020, which further influences the local dust? If the relationship between BC and DU is identified during COVID-19, does it still hold in the long-term statistics? COVID-19 induced emission reductions offer a unique

opportunity to disentangle the fast response of the climate system to strong and abrupt changes in anthropogenic aerosols.

In this work, we first show that Indian DU is positively correlated to northern Indian BC while negatively correlated to southern Indian BC based on a long-term statistical analysis. Then we identify the mechanism of BC-climate interactions regulating Indian DU revealed during COVID-19 based on satellite and ground-based observations and model simulations. Consequently, the outbreak of the subsequent Indian summer monsoon is delayed.

## Results

**Dipole pattern of BC-DU relation and resulting DU changes in long-term statistics and during COVID-19.** To examine whether BC and DU change concurrently in the premonsoon season, a long-term time series of BC and DU burdens from 2000 to 2020 is applied from the Modern-Era Retrospective analysis for Research and Applications version 2 (MERRA-2) Reanalysis. We analyze the correlation coefficient of anomalous time series (relative to monthly means) between DU over northern India (averaged over 25°–35°N, 70°–88°E) and BC at each grid cell over the domain covering the Indian subcontinent in April-May. A dipole pattern of correlation is found with northern Indian BC positively correlated and southern Indian BC negatively correlated to the northern Indian DU. The correlations peak over northwestern and southwestern India and can reach up to ±0.6 (Fig. 1b). We further decompose the time evolution of averaged DU over northern India into averaged BC over northern and southern India during the premonsoon season using multiple linear regression, and obtain the following equation:

$$\triangle DU_{north} = 0.56 \times \triangle BC_{north} - 0.55 \times \triangle BC_{south} + \varepsilon \quad (1)$$

where $\triangle DU_{north}$ is the DU anomaly averaged over northern India, $\triangle BC_{north}$ is the BC anomaly averaged over northern India (25°–35°N, 70°–88°E), $\triangle BC_{south}$ is the BC anomaly averaged over southern India (15°–25°N, 70°–88°E), and $\varepsilon$ is the residual. It shows that $\triangle DU_{north}$ is well correlated ($r = 0.75$) with the sum of $\triangle BC_{north}$ and $\triangle BC_{south}$. As seen from the two regression coefficients, the roles of northern and southern Indian BC in regulating the northern Indian DU are comparable. We note that neither $\triangle BC_{north}$ or $\triangle BC_{south}$ alone can explain $\triangle DU_{north}$ ($r = 0.5$) as well as their combination, implying that northern and southern Indian BC work together to facilitate Indian DU changes.

During COVID-19, the Indian industrial zone located in the IGP experiences the maximum nationwide reduction of anthropogenic emissions, as denoted by nitrogen dioxide ($NO_2$) from the TROPOspheric Monitoring Instrument satellite (TROPOMI) (Fig. 1c). The reduction of BC over northern India[33–35] results from reduced emissions from sectors such as industry, transport[26,27], and biomass burning (Supplementary Fig. 1). Meanwhile, the southern Indian BC burden increases (Fig. 1d) in April-May 2020 due to the intensified CRB (Supplementary Fig. 1a). Following the relationship of DU and BC shown in Eq. (1), a record low (exceeding 1.5 standard deviations) of dust optical depth (DOD) over India observed by the CALIPSO (starting from 2007) occurs in April-May 2020 especially over northern India with evidence from changes in the CALIPSO DOD and Aerosol Robotic Network (AERONET) coarse-mode AOD (Fig. 1a, e). From the analysis of the DU budget over northern India based on the MERRA-2 Reanalysis, the reduction of northern Indian DU is mainly a result of weakened DU transport across the west boundary, enhanced DU transport across the south boundary away from northern India (see below for wind changes) and the suppressed local DU emission (Supplementary Fig. 2, see below for reasons). Owing to the reductions of both natural DU burden and

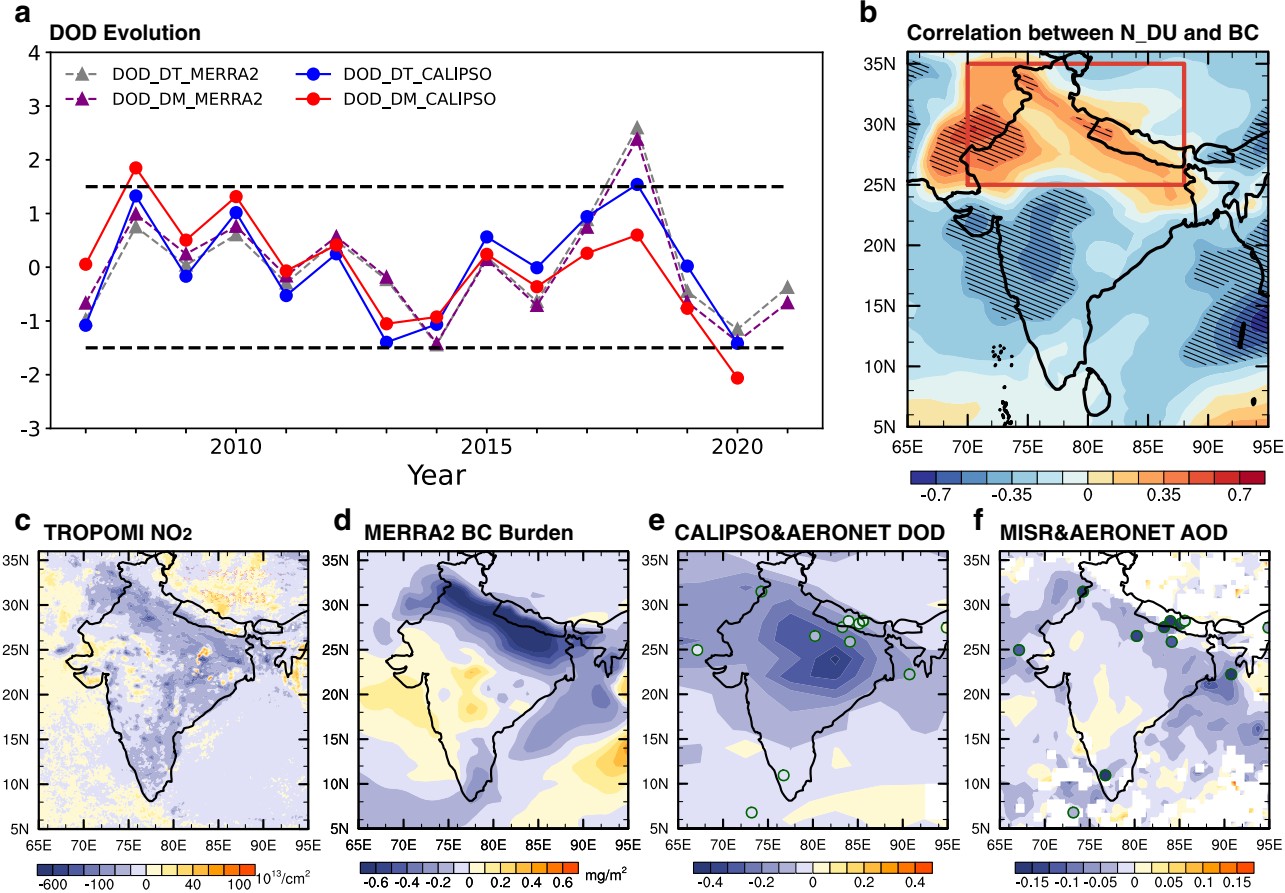

**Fig. 1 BC, DU, AOD, and NO₂ for the premonsoon season. a** Time series of normalized anomalies of dust optical depth (DOD) after removing the long-term trend (DOD_DT, blue) and after subtracting the long-term mean (DOD_DM, red) over India from CALIPSO (2007–2020) (solid lines) as well as the counterparts (DOD_DT in gray and DOD_DM in purple) from MERRA-2 (2007–2021) (dashed lines). **b** Correlation coefficients between northern Indian DU (averaged over the square) and BC over each grid cell of the Indian subcontinent from MERRA-2 in April-May. Areas exceeding the 90% confidence level are hatched using Student's $t$-test. **c**–**f** Spatial distributions of differences of **c** NO₂ concentrations from TROPOMI, **d** BC burdens in MERRA-2, **e** AOD of dust in CALIPSO (contour fill) and of coarse-mode in AERONET (colored circles), and **f** aerosol optical depth (AOD at 550 nm) from MISR (contour fill) and AERONET (colored circles) between 2020 and 2015–2019.

anthropogenic emissions, a pronounced decrease in aerosol optical depth (AOD) in northern India is consistently observed by the Multi-angle Imaging SpectroRadiometer (MISR), AERONET, Moderate-resolution Imaging Spectroradiometer (MODIS), and MERRA-2 (Fig. 1f and Supplementary Fig. 3). However, AOD in southern India is only slightly changed due to the increase in biomass burning emissions there.

As a comparison, the BC burden in April-May of 2021 without the COVID-19 lockdown increases over the Indian subcontinent compared to that of 2020 and is even larger than the 2015–2019 climatological mean (Supplementary Fig. 4a, c). As predicted by Eq. (1), DU over northern India in April-May 2021 begins climbing as seen from MERRA-2 (Fig. 1a and Supplementary Fig. 4b). Since there are still positive BC anomalies over southern India, the negative anomalies of northern Indian DU relative to the 2015–2019 climatology remain (Supplementary Fig. 4d).

**DU response to BC climate impacts identified during COVID-19.** The convolution of BC and DU changes during the pre-monsoon season is unveiled with the COVID-19 pandemic. The mechanisms of how the dipole pattern of BC-DU correlation and the contrasting changes of northern and southern Indian BC affect the dust burden in India are explored and identified below. As absorbing aerosols, BC heats the atmosphere[11,13,14]. As a result, the shortwave heating rate in the atmosphere is decreased

in northern India when BC emissions decrease during the pandemic. The reduction of heating is mainly located in the lower troposphere of northern India and can extend to 600 hPa (Fig. 2a). With the decrease of BC in the atmosphere, the deposition of BC on snow over the southern slope of the TP and the Himalayas is reduced, leading to an increase of surface albedo (Fig. 2b)[15–18]. Enhanced surface albedo reflects more solar radiation, thereby cooling the surface. The combination of these two BC effects cools the air temperature, especially in the lower troposphere, and leads to maximized cooling near the surface along the southern TP (Fig. 2c)[36,37]. The cooling in the atmospheric column induces a strong descending motion between 25°N to 30°N, peaking near 30°N. Near the surface, colder and heavier air accumulated near the southern slope of the TP drains southward and cools northern India (Supplementary Fig. 5a, c).

The atmospheric stability, defined here as the potential temperature difference between 700 and 900 hPa[38], increases in northern India (Supplementary Fig. 5b). This favors the development of low clouds there[38] (Fig. 2d). The increase in low clouds is also consistent with the adiabatic cooling of warmer and more moist air in the planetary boundary layer (PBL) in India lifted isentropically by the cold air from TP (Supplementary Fig. 5d). In comparison with northern India, BC increases from intensified CRB in southern India heat the lower atmosphere and decrease the atmospheric stability (Fig. 2a and Supplementary

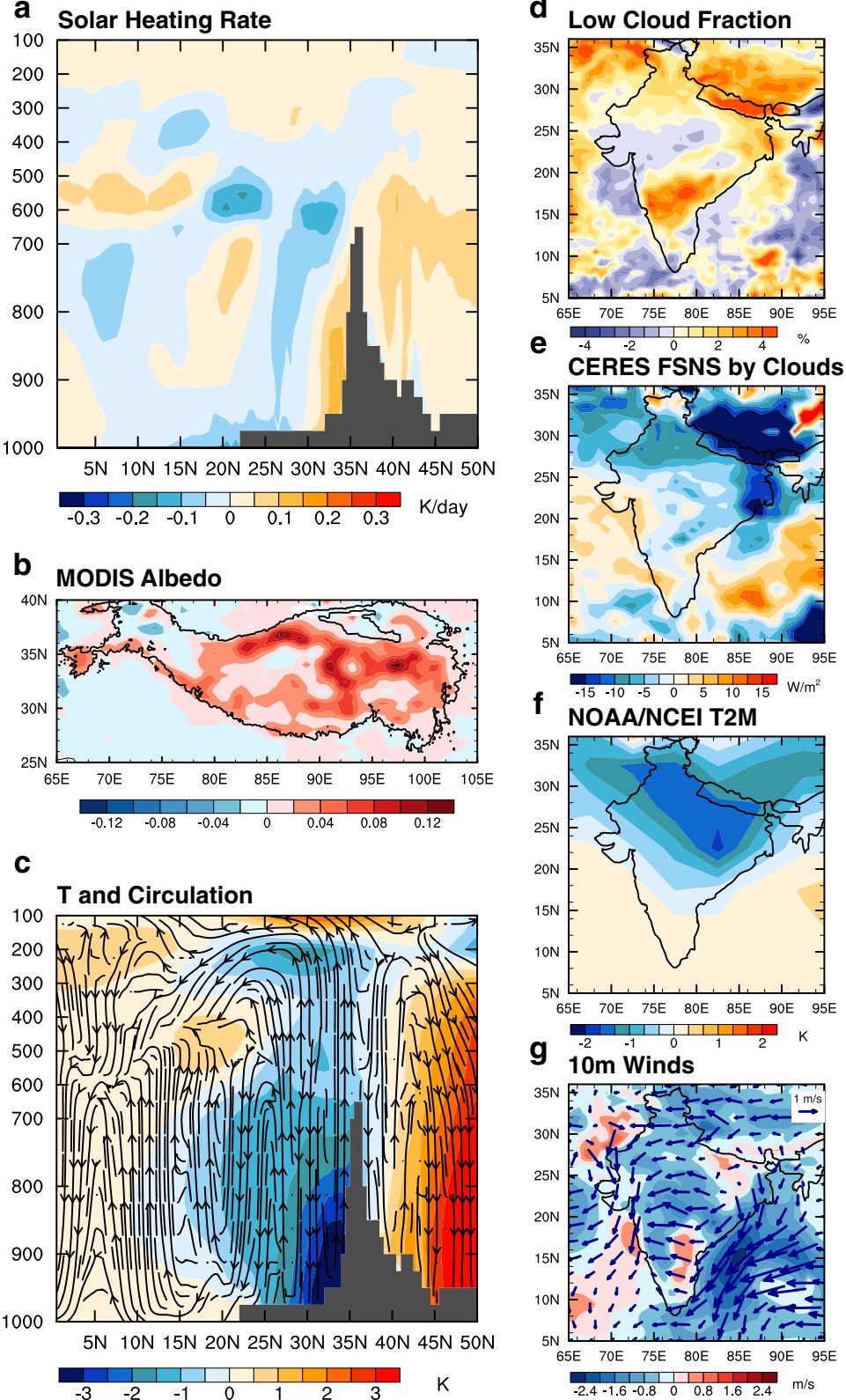

**Fig. 2 BC induced climate anomalies in the premonsoon season of 2020 during the COVID-19 pandemic. a** Zonal mean (70°E–90°E) shortwave heating rate in MERRA-2. **b** Surface albedo in MODIS. **c** Zonal mean (70°E–90°E) air temperature (contour fill) and stream function (vectors) in MERRA-2. **d** Low cloud fraction in MERRA-2. **e** Surface shortwave cloud radiative effect in CERES. **f** 2-m air temperature in NOAA/NCEI. **g** Winds at 10 m in MERRA-2. All anomalies are relative to 2015–2019 multiyear means.

Fig. 5b), which is unfavorable for the low cloud formation (Fig. 2d). The net surface shortwave radiation over northern and eastern India is reduced (Fig. 2e and Supplementary Fig. 6a, c) as observed from the Clouds and the Earth's Radiant Energy System (CERES), which is consistent with the increase in low clouds, although net shortwave radiation by aerosols at the surface (see Methods) is actually increased there because of the BC emission reduction (Supplementary Fig. 6b, d). Over southern India, net shortwave radiation by aerosols at the surface slightly decreases (Supplementary Fig. 6d) due to elevated CRB emissions. Reduced net surface shortwave radiation together with cold air advection by anomalous southward winds from the TP (Supplementary Fig. 5c), cools the surface air temperature by up to 2 K over northern India, as observed by the National Oceanic and Atmospheric Administration/National Centers for Environmental Information (NOAA/NCEI) (Fig. 2f). Therefore, a positive sea-level pressure (SLP) anomaly emerges over northern India (Supplementary Fig. 7c), directly coinciding with the area of maximum column cooling (Fig. 2c). The resulting anomalous easterly surface wind on the southern end of the anticyclonic anomaly slows down the climatological westerly wind (Fig. 2g).

In comparison with negative northern Indian BC anomalies, the positive southern Indian BC anomalies are equally important for establishing the meridionally asymmetric cooling, which contributes to the easterly wind anomalies. The cooling over northern India resulting from BC reduction and warming over southern India from increased CRB BC emissions induce an anomalous southward pressure gradient force. To satisfy the geostrophic approximation, especially at levels above the near-surface, this must be balanced by an anomalous northward Coriolis force as the emergence of easterly wind anomalies. Owing to anomalous easterly wind, the eastward transport of dust from the Middle East and Sahara as well as local dust emissions in the Thar Desert are suppressed (Supplementary Figs. 7b, 2a), leading to a record low of dust loading in 2020. The distinct roles of northern and southern BC in regulating the northern Indian DU show a cutoff along ~25°N, which is consistent with the southern boundary of the westerly dust transport belt (Fig. 1b and Supplementary Fig. 7a). This is formed by the opposite anomalous meridional pressure gradient forces induced by BC radiative heating in northern and southern India.

**Evidence from model simulation.** To corroborate our findings of the effect of the contrasting anomalies of northern and southern Indian BC on DU in India, we conduct climate simulations using the Community Atmosphere Model version 6 with chemistry (CAM6-chem) in the NCAR Community Earth System Model Version 2 (CESM2). Two experiments with different emission datasets, namely SSP and COVID-19, are designed. In the COVID-19 case, to represent the decreases and increases of BC emissions in northern and southern India respectively, we only use the Forster COVID-19 BC emission inventories[25] in northern India while using 120% of SSP245 BC emission inventories in southern India (see Methods). By contrasting these two experiments, we intend to reproduce the opposite BC changes in northern and southern India as discussed before; therefore, the difference in the thermodynamic and aerosol fields between the two cases will be interpreted in a similar manner as our observation/MERRA-2 data analyses. In addition to these two cases, in order to elucidate the relative importance of solar heating in the atmospheric column and snow-darkening effect (SDE), we designed another case named NOSDE, in which the same emission dataset as the COVID-19 case is used but the SDE in the Snow, Ice, and Aerosol Radiative (SNICAR) model is turned off. For each experiment, we conduct 20 ensemble members of

climate simulations, each of which is differentiated by a small and unique temperature perturbation (see Methods). Overall, the CAM6-chem model successfully reproduces the observed meteorological and aerosol fields in 2015–2019 as shown in Supplementary Figs. 8–15 (e.g., AOD, DOD, BC, surface albedo, clouds, precipitation, and dynamic fields).

Figure 3a shows the difference in BC burdens in April and May between the COVID-19 case and the SSP case. A significant reduction of BC burden can be seen in northern India, while a moderate increase of BC burden can be seen in southern India. This pattern as well as the magnitude of the BC difference is in good agreement with the MERRA-2 Reanalysis (Fig. 1d). Not surprisingly, the BC reduction peaks over the IGP because of the pandemic's lockdown effect. Comparing the COVID-19 case to the SSP case, we find that less BC in the atmosphere causes cooling in the atmospheric column by a magnitude of ~−1 K from 25°N to 30°N and a descending motion over the same region at pressures >~600 hPa (Supplementary Fig. 16a). In alignment with the MERRA-2 Reanalysis, we find that the descending motion can enhance the atmospheric stability and trigger a chain of dynamic responses in northern India: enhanced low cloud fraction by up to 2% (more in northern and eastern India and less in western India, Fig. 3b), reduced surface temperature by 1–1.5 K, and higher SLP by 0.8 hPa (Supplementary Fig. 16b, c). The model results exhibit some notable differences compared to the MERRA-2. For instance, the changes in the dynamic fields between two model experiments are smaller compared to the MERRA-2 results in terms of magnitude. This can be contributed by many factors, including weaker surface albedo changes over the TP (Supplementary Fig. 16d in comparison with Fig. 2b), other absorbing aerosols such as brown carbon from the CRB not considered in the model, and different model years from the MERRA-2 Reanalysis (see Methods). In addition, the model predicts a slightly higher low cloud fraction increase over southern India where BC increases (Fig. 3b). We speculate that this might be due to more moisture transport from Bengal Bay triggered by the BC opposite changes in northern and southern India (Fig. 3c).

Similar to the MERRA-2 results, the COVID-19 experiment successfully predicts easterly wind anomalies over the Indian continent and the adjacent oceans (Fig. 3c), leading to a strong reduction in the DU burden (Fig. 3d). The strongest DU reduction signal stretches along the IGP. The bar chart imposed on Fig. 3d shows the difference in the DU budget between the two cases. Similar to the MERRA-2 results, two major causes for the DU reduction are weaker zonal transport ($-0.41$ Tg month$^{-1}$) and local dust emission ($-0.31$ Tg month$^{-1}$). Because the COVID-19 case predicts a southerly wind anomaly in southern India, the meridional transport in the model actually contributes to the increase in DU burden.

Figure 3e shows that the reduced BC in northern India (>25°N) significantly cools the atmosphere column from surface to about 600 hPa with a magnitude as strong as $-0.06$ K day$^{-1}$. In southern India, we see a small warming effect because BC emission is increased in this region by 20%. The cooling effect associated with the dust reduction has a smaller magnitude at about $-0.02$ to $-0.03$ K day$^{-1}$ (Fig. 3f), particularly above 900 hPa level, despite the reduction in dust burden is about two orders of magnitude larger than that in the BC burden. This is because of the much stronger mass absorption efficiency of BC aerosol.

The comparison between the COVID-19 and NOSDE cases shown in Supplementary Fig. 17a indicates that the SDE of BC accounts for about one-tenth of easterly wind anomalies. As shown in Supplementary Fig. 17b, the SDE roughly accounts for less than 5% of the DU reduction in the northern part of India. In the central part of India (between 15°N and 20°N), the SDE contributes to as

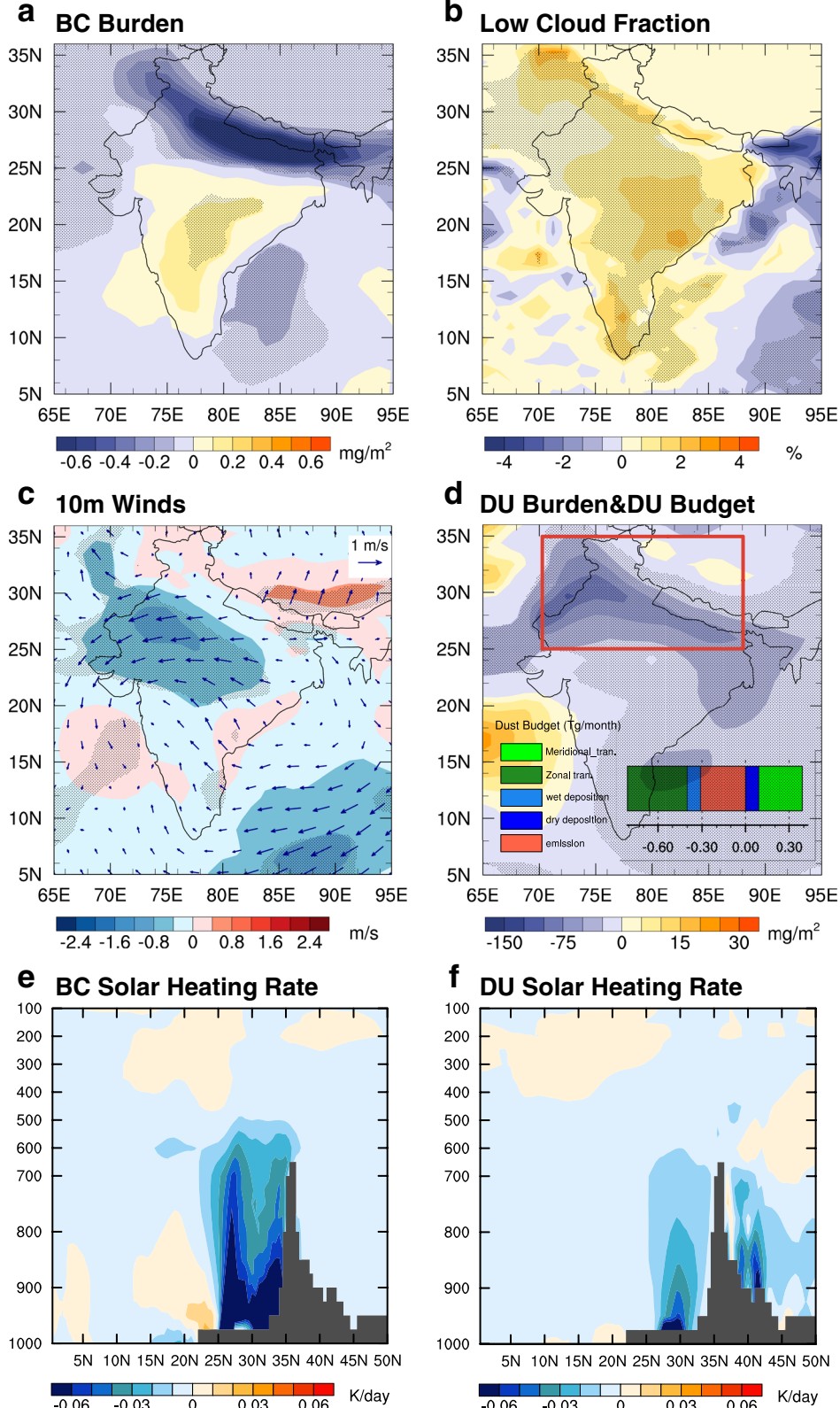

**Fig. 3 BC-reduction induced climate anomalies in April and May modeled by the COVID-19 case in comparison with the SSP case. a** BC burden. **b** Low cloud fraction. **c** Winds at 10 m. **d** Dust burden. The solar heating rate associated with **e** BC and **f** Dust. The bar chart imposed in (**d**) shows the difference in modeled dust budget between two cases. Differences with a confidence level greater than 90% are stippled using the Student's *t*-test and Monte Carlo field significance test.

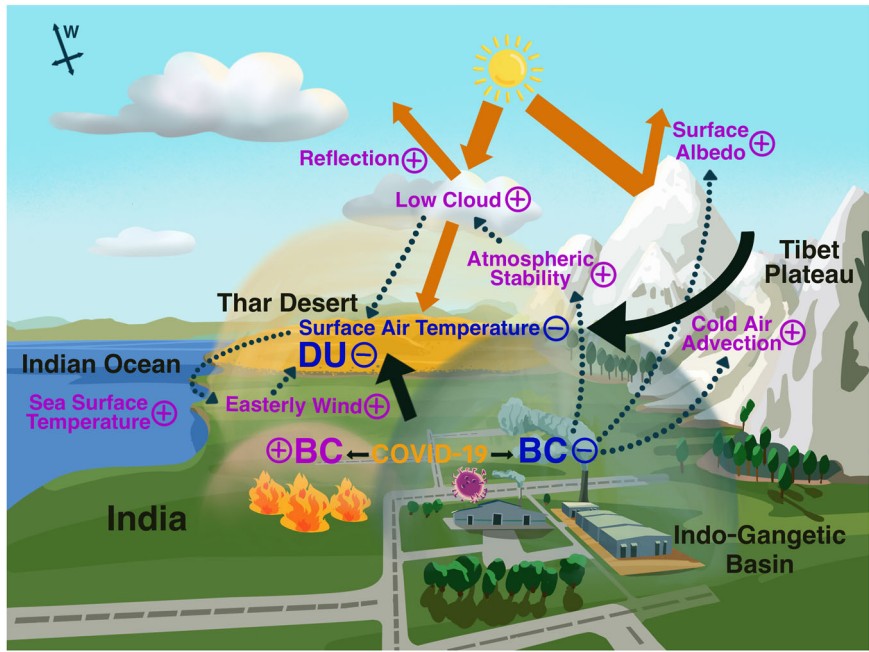

**Fig. 4 Schematic interpreting the impacts of northern Indian BC reduction due to lockdown and southern Indian BC increase due to intensified crop residue burning during the Indian premonsoon season for the COVID-19 pandemic case.** With the reductions of northern Indian BC and DU, the subsequent Indian summer monsoon outbreak is delayed. Note that the proposed mechanisms work not only for the COVID-19 case but also is valid in the long-term statistics.

high as 20% of the DU reduction. Averaged over entire India, SDE contributes to 4% of the DU reduction. Therefore, the reduced solar heating in the atmospheric column due to BC reduction is the major factor triggering the DU reduction.

## Discussion

The BC-induced climate change mechanisms are highlighted in Fig. 4, along with a mathematical description of the circulation change (see Methods). BC reduction over northern India due to the COVID-19 pandemic decreases solar heating in the atmospheric column and increases the surface albedo of the TP due to the reduced BC snow-darkening effect. The atmosphere responds to this cooling with a descending motion, and the cooling is maximized near the surface across the southern TP, as the higher surface albedo accentuates the cooling. As a result, the colder heavier air from TP slips downhill into the IGP (a katabatic flow), and the warmer and moister IGP air is lifted isentropically over the cold air, forming low clouds. The atmospheric stability is enhanced, which also favors the formation of low clouds, resulting in more reflection of solar radiation by clouds. Together with cold air advection, less solar radiation at the surface reduces surface air temperature. The regional cooling and sinking motion triggers anomalous high sea-level pressure. This, along with meridionally asymmetric cooling induced by negative BC anomalies over northern India and positive BC anomalies over southern India, results in the anomalous easterly wind, weakening the westerly low-level jet. Hence, dust transport from the Middle East and Sahara as well as dust emissions from the Thar Desert are suppressed. With the proposed mechanisms, we can see the vital roles of BC in northern and southern India in regulating the Indian DU during the pre-monsoon season through the dipole pattern of BC-DU correlation. The long-term snow cover over the southern slope of the TP provides the precondition for the BC snow-darkening effect. We note that the rainfall anomalies observed by the Global Precipitation Measurement (GPM) are small (Supplementary Fig. 19a) in April-May 2020 and thus have a negligible effect on the DU burden.

Previous studies[11,13] have suggested the effects of aerosol emission increases on the Indian climate during the premonsoon and summer monsoon periods, and most of them were based on the analyses of modeling results. In this study, we show clear evidence of aerosol effects based on the analyses of observation and reanalysis data, and these effects exhibit the opposite pattern to previous studies when considering pandemic-related emission reductions: cooling of the atmospheric layer, descending motion, and an easterly wind anomaly. In addition, this study reveals for the first time the dipole pattern of BC-DU correlation in regulating the Indian DU in the pre-monsoon season. The BC anomalies in southern India have a contrasting effect on DU as that of the northern Indian BC anomalies. Since DU, like BC, is an absorbing aerosol, dust decreases induced by the decreases (increases) of BC in northern (southern) India could further lead to a cooler and cloudier April-May across northern India. We note that the mechanisms identified in this study for the BC-induced premonsoon climate change during COVID-19 can also be applied to other years (Fig. 1a, b). As an example, India experienced substantially high dust loadings during April-May 2018 (Fig. 1a)[39]. Overall, the BC anomalies and induced climate changes are opposite to those in the COVID-19 case. As shown in Supplementary Fig. 18, both DU and BC increased in northern India compared with the 2015–2019 means (Supplementary Fig. 18a–c). Higher BC loading strengthened the solar heating in the atmospheric column at 20–30°N (Supplementary Fig. 18d), and the anomalous BC decreased the snow albedo over the TP and the Himalayas (Supplementary Fig. 18e)[15,18]. Low clouds were reduced, and solar radiation reaching the surface was increased accordingly (Supplementary Fig. 18h, i). As a result, the near-surface air temperature increased, especially over western India (Supplementary Fig. 18j). Under these circumstances, the strengthened dust transport flux from the Middle East and Sahara contributes to the increase in Indian DU (Supplementary Fig. 18g).

With negative BC and DU anomalies in northern India during the premonsoon season, a delay of the Indian summer monsoon outbreak in 2020 is found (Supplementary Fig. 19c) (see

Methods). As a result, decreased precipitation emerges over northern India in the following months (Supplementary Fig. 19b, c)[13], which results from the declined flow of moisture from the Indian Ocean associated with weakened large-scale advection (Supplementary Fig. 20). Owing to the reduction in BC and DU radiative effects, it cools the atmosphere and triggers anomalous anticyclonic circulation in the lower troposphere, which restrains the convergence of water vapor from the Indian Ocean. This mechanism has been proposed as the elevated heat pump hypothesis[13–17] but with the opposite sign in this study. The findings in this study imply that as anthropogenic aerosol and precursor emissions in northern India decline in the future, regional dust loading will decrease through the BC-climate interactions. The reductions in BC and DU in India may also contribute to the weakening of the Indian summer monsoon in the future. This indicates co-benefits of BC reduction for air quality, human health, and climate change[5,40,41].

## Methods

**Observation and reanalysis data**. The Modern-Era Retrospective Analysis for Research and Applications, Version 2 (MERRA-2) Reanalysis is used in the analyses of BC and DU burdens, AOD, and DOD, which has a horizontal resolution of $0.5° \times 0.625°$ and a vertical resolution of 42 levels from the surface to 0.1 hPa[42]. In addition to MERRA-2 AOD and DOD, AOD from the Multi-angle Imaging Spectro-Radiometer (MISR, $0.5° \times 0.5°$) level3 aerosol product (MIL3MAEN_4)[43], the Moderate-Resolution Imaging Spectroradiometer (MODIS, $1° \times 1°$) Aqua level-3 product (MYD08_M3)[44], and the AErosol RObotic NETwork (AERONET) observations[45] are also adopted in the analyses. Additional observation data used in this study include the DOD approximately obtained by the coarse-mode extraction from AERONET observations and directly observed by the Cloud-Aerosol Lidar and Infrared Pathfinder Satellite Observations (CALIPSO, $2° \times 5°$) level-3 aerosol product (cloud-free/clear sky, V4-20)[46] as well. Surface $NO_2$ concentrations are from TROPOspheric Monitoring Instrument (TROPOMI, $0.125° \times 0.125°$)[47]. BC emissions from anthropogenic sectors are given by ref. [26,27] and those from biomass burning are based on the Global Fire Emissions Database version 4 (GFED4.1 s)[48] (Supplementary Fig. 1).

As for non-pollution fields, the following observation and reanalysis data are used in this study: the MODIS daily product (MCD43C3 v006)[49] with a resolution of $0.05° \times 0.05°$ for surface albedo, the Clouds and the Earth's Radiant Energy System (CERES) level-3b EBAF datasets[50] with a resolution of $1° \times 1°$ for surface radiation, the National Oceanic and Atmospheric Administration/National Centers for Environmental Information (NOAA/NCEI, $5° \times 5°$)[51] for 2 m air temperature, and the Global Precipitation Measurement observations (GPM, $0.5° \times 0.625°$)[52] as well as MERRA-2 for rainfall. The remaining fields (e.g., low clouds, winds, and air temperature) are all from MERRA-2.

**Time series analysis**. DOD in MERRA-2 from 2007 to 2021 as well as in CALIPSO observations from 2007 to 2020 in April-May are detrended first, retaining their interannual variabilities. The DODs in the MERRA-2 and CALIPSO observations subtracted by 2007–2021 and 2007–2020 means are calculated, respectively. For a demonstration of these anomalies together in one figure, they are all normalized (Fig. 1a). With detrended MERRA-2 BC and DU burden anomalies in April-May from 2000 to 2020, the correlation coefficients between DU anomalies averaged over northern India ($25°–35°N$, $70°–88°E$) and BC burdens over each MERRA-2 grid for all 21 years are calculated.

**Shortwave radiative forcing due to aerosols and clouds**. In CERES observations, the shortwave radiative forcing at the surface perturbed by aerosol changes during the COVID-19 pandemic approximates $\Delta F_{clear}$, where $\Delta$ is the difference between 2020 and 2015–2019 mean and $F_{clear}$ is clear-sky net shortwave radiative flux at the surface. That is due to changes of clouds approximates $\Delta(F-F_{clear})$, where $F$ is all-sky net shortwave radiative flux at the surface. Different from observations, MERRA-2 and CESM2 can diagnose surface radiative forcing due to aerosols and clouds cleanly. In them, the radiative forcing due to aerosols is estimated as $\Delta(F-F_{clean})$, where $F_{clean}$ is a diagnostic net shortwave radiative flux with neglecting the absorption and scattering of radiation by aerosols. Radiative forcing due to clouds is calculated as $\Delta(F_{clean}-F_{clean,clear})$, where $F_{clean,clear}$ is an additional diagnostic net radiative flux with neglecting the absorption and scattering of radiation by both aerosols and clouds.

**Indian summer monsoon outbreak**. To identify the onset time of the Indian summer monsoon outbreak over northern India, a method similar to[53] is applied. GPM daily precipitation at each grid over northern India from May to August in 2015–2020 is spatially averaged. The time series of northern Indian precipitation is subtracted by its climatological mean (averaged over 2015–2019) in May. Then a 1-

2-1 filter is used to smooth the anomalies. The date when the anomaly changes sign from negative to positive (i.e., precipitation is above the climatological mean of May) and the anomalies in the following days are consecutively positive is referred to as the onset of the rainy season.

**Model configuration**. We used the Community Atmosphere Model version 6 with chemistry (CAM6-chem) in the Community Earth System Model version 2.0 (CESM2) for the simulation. In CAM6-chem, the four-mode version of the Modal Aerosol Module (MAM4)[54] coupled with the MOSAIC (Model for Simulating Aerosol Interactions and Chemistry) scheme is used to simulate the aerosol microphysics and chemistry[55]. In this MAM4-MOSAIC configuration, a gas-aerosol exchange is simulated by MOSAIC, while the other aerosol processes, like coagulation, dry/wet deposition, and renaming are handled by MAM4. The default MAM4 treats aerosol species of sulfate, mineral dust, sea salt, black carbon (BC), primary particulate organic matter (POM), and secondary organic carbon (SOA) in the Aitken, accumulation, coarse, and primary carbon modes. BC and POM from anthropogenic and biomass burning sources are emitted to the primary carbon mode. After the aging process, BC and POM are transferred to the accumulation mode. MAM4-MOSAIC additionally treats nitrate and ammonium aerosols. 5% of mineral dust aerosols are considered as calcium carbonate ($CaCO_3$) and sea salt aerosols are partitioned into sodium ($Na^+$), chloride ($Cl^-$), and a small fraction of sea salt sulfate, so that the processes of $NO_3^-$ replacing $CO_3^{2-}$ in $CaCO_3$ and $Cl^-$ in NaCl are treated. Several chemical reactions, such as $O_3$-$NO_x$-$HO_x$ chemistry and $N_2O_5$ hydrolysis, need to be considered to drive MOSAIC; therefore, the MOZART-TS1 (Model for Ozone and Related chemical Tracers, troposphere, and stratosphere) full chemistry mechanism is turned on in CAM6-chem[56].

The MAM4 is coupled with the RRTMG radiation package[57] and the two-moment cloud microphysics scheme[58] to enable the aerosol-radiation and aerosol-cloud interaction calculations. In CESM2, boundary layer turbulence, shallow convection, and cloud macrophysics are treated by the Cloud Layers Unified by Binormals scheme (CLUBB)[59–61]. The snow-darkening effect due to BC and DU deposition on snow is considered by the Snow, Ice, and Aerosol Radiative (SNICAR) model[62].

CESM2 is run with the standard resolution (~1° horizontal resolution, 32 levels to 3 hPa, model time step of 1800 s) and "FC2010climo" component set ("F" indicates the atmosphere and land components of CESM2 are active, while the sea ice and sea surface temperatures (SST) are prescribed. "C" indicates the MOZART-TS1 full chemistry is turned on. "2010climo" indicates that 2010 climatological SST and sea ice are used). To examine the effect of BC anomalies on Indian climate and resulting changes in the dust field, we conduct three sets of experiments, namely "SSP", "COVID-19", and "NOSDE". In the SSP experiment, we use the SSP245 emission of 2020, while in the COVID-19 case, BC emission of 2020 over India from Forster et al. (2020) is used. We found that the anthropogenic BC emission is reduced in India according to Forster et al. (2020); however, according to MERRA-2 Reanalysis, this is not the case in southern India (<25°N), where BC burden in 2020 is slightly higher compared to previous years (partially due to increased agricultural fires in this region). Consequently, we only use the Forster BC emission in northern India, while we use 120% of SSP245 BC emission in southern India. We use $1.95 - 0.79i$ for the refractive index of BC in the model following ref. [2]. The configuration of NOSDE is identical to the Covid-19 case, but the snow-darkening effect (SDE) in the SNICAR model is turned off.

In addition, to better simulate the dust fields over India in our study, we adopt the Kok dust emission scheme[63] instead of the default Zender dust emission scheme[64] in CESM2. By comparing the performance of two different dust emission schemes, we find that the Zender scheme significantly underestimates the DOD in India in April and May by a factor of three. In contrast, the Kok scheme reasonably reproduces DOD in India during the same period.

The Kok scheme is developed on the basis that sandblasting is the main mechanism for dust emission. The scheme accounts for two processes missing in most existing parameterizations: a soil's increased ability to produce dust under saltation bombardment as it becomes more erodible, and the increased scaling of the dust flux with wind speed as the soil becomes less erodible. The dust emission flux (F) is then calculated from the friction velocity, threshold friction velocity, atmospheric density, clay content in the soil, and the areal fraction of exposed bare soil. Specifically, F is proportional to the n-th power of surface wind strength (defined as the friction velocity) and $n = \alpha + 2$ where $\alpha$ (>0) is proportional to the relative erodibility of the surface. The Kok scheme showed significant improvements over the Zender scheme used in the default CESM2[65].

The emitted dust particles are partitioned into Aitken, accumulation, and coarse modes following the dust size distribution of ref. [66], which is derived from the assumption that breaking of soil particles during dust emissions is analogous to the fragmentation of brittle materials. The size ranges for the three dust modes in diameter are 0.01–0.1, 0.1–1.0, and 1.0–10 μm, respectively and according to ref. [66], the mass fractions for the three dust modes are 0.00165, 1.1, and 99.9%, respectively. Dust deposition includes both dry and wet deposition, which are treated separately. For dry deposition, CESM2 adopts the deposition of ref. [67]. For wet deposition, the in-cloud and below-cloud scavenging are treated separately, following the parameterizations described in ref. [68].

With the aforementioned configuration, we conduct 20-member ensemble simulations for both experiments. The ensemble members are differentiated by a

small perturbation in temperature. For each member, we conduct a two-year simulation with the second-year simulation used for model analysis and the first-year simulation used as model spin-up.

**Model performance.** The model performance for April-May in India was examined thoroughly. The simulated AOD, BC burden, low clouds, liquid water path, atmospheric circulation, shortwave heating rate, air temperature, the surface albedo of the TP, and precipitation are comparable with observations and/or MERRA-2 (Supplementary Figs. 8–15). After using the Kok dust emission scheme, modeled DOD matches well compared to MISR DOD over the Indian source region (Supplementary Fig. 9a vs. Fig. 9c); however, the magnitude is underestimated over the transported route. The discrepancy can be possibly contributed to many factors, for example, biased assumption of dust particle size and/or overestimated wet and dry deposition[69,70]. Since we explored the dust change due to BC reduction rather than the absolute value, the underestimation of the simulated DOD has small influence on the fidelity of the model performance.

**Horizontal temperature advection and adiabatic heating/cooling.** Horizontal temperature advection can be written as $-\overline{\mathbf{V}_h \bullet \nabla_h T}$, where overbar denotes the time mean, $\mathbf{V}_h$ is horizontal wind vector and $T$ is air temperature. Adiabatic heating (downward motion) or cooling (upward motion) can be represented by $\overline{S_p \omega}$ where $\omega$ vertical velocity in pressure ($p$) coordinates and $S_p$, the stability parameter, is defined by $S_p \equiv -\frac{T}{\theta}\frac{\partial \theta}{\partial p}$ ($\theta$ is potential temperature). Calculations are based on 3-hourly data from MERRA-2, as shown in Supplementary Fig. 5c, d.

**Mathematical derivation of circulation change due to BC.** The circulation changes can be described using the concept of thermal vorticity ($\zeta_T$), defined to be the difference of the geostrophic vorticity between two pressure levels in a flat, non-accelerating, and non-viscous flow:

$$\zeta_T \equiv \zeta_U - \zeta_L \tag{2}$$

where the subscripts $U$ and $L$ denote an "upper-" and "lower-" level isobaric surface within a given column of air. Geostrophic vorticity $\zeta$ is proportional to the Laplacian of the geopotential height field only in an atmosphere subject to the $\beta$-plane approximation:

$$\zeta = \frac{1}{f_0}\nabla_p^2 \Phi \tag{3}$$

where $f_0$ is a constant Coriolis parameter, $\Phi = gz$. Where, $g$ is the gravitational acceleration, and the $p$ denotes that the Laplacian is taken along a constant-pressure surface. Differentiating Eq. (3) with respect to time while using Eq. (2), we find,

$$\frac{\partial \zeta_T}{\partial t} = \frac{1}{f_0}\nabla_p^2 \frac{\partial \triangle \Phi}{\partial t} \tag{4}$$

where $\triangle \Phi$ is the layer thickness between two isobaric surfaces. Since the local tendency in $\triangle \Phi$ is directly proportional to the column-averaged temperature via the hypsometric equation (not presented), it can be shown using Eq. (4) that, assuming waveform solutions, the column temperature change and resulting layer thickness tendency of opposing sign to the thermal vorticity tendency. That is, for a positive (negative) layer thickness tendency, the thermal vorticity tendency is negative (positive).

Upon ascertaining the sign of $\frac{\partial \zeta_T}{\partial t}$, the induced vertical motion response can be estimated by considering the simple geostrophic vorticity tendency at each level:

$$\frac{\partial \zeta}{\partial t} = -\delta f_0 = -\left(\frac{\partial u}{\partial x}+\frac{\partial v}{\partial y}\right)f_0 = \frac{\partial w}{\partial z}f_0 \tag{5}$$

where $w$ is the vertical velocity in height coordinate. Using Eq. (5), the local vorticity tendency at a level can be related to the divergence ($\delta$), whose vertical derivative is proportional to the Laplacian of $w$:

$$\frac{\partial \delta}{\partial z} = -\nabla_{x,y}^2 w \tag{6}$$

Pandemic-related BC reductions in April-May 2020 compared to other years (Fig. 1d) led to decreases in the column solar heating rate (Fig. 2a) and column cooling between the surface and 600 hPa (Fig. 2c). Layer cooling leads to concurrent layer thickness reductions $\left(\frac{\partial \triangle \Phi}{\partial t} < 0\right)$, so $\frac{\partial \zeta_T}{\partial t} > 0$, meaning $\frac{\partial \zeta}{\partial t}\big|_U > \frac{\partial \zeta}{\partial t}\big|_L$ by Eq. (4). By Eq. (5), these thermodynamic perturbations lead to cyclonic and convergent (anticyclonic and divergent) vorticity spin-up on the "upper-level" ("lower-level") isobaric surface. By Eq. (6), $\frac{\partial \delta}{\partial z} < 0$, and $-\frac{\partial \delta}{\partial z} > 0$, making $w < 0$ (sinking vertical motion; Fig. 2c) assuming waveform solutions.

## Data availability

MERRA-2 data is available from https://disc.sci.gsfc.nasa.gov/datasets/. CERES radiation data can be accessed online at https://ceres.larc.nasa.gov/. CALIPSO, MODIS, and MISR data are available from https://www-calipso.larc.nasa.gov, https://ladsweb.modaps.eosdis.nasa.gov/, and https://asdc.larc.nasa.gov/-data/MISR/. AERONET data is available at https://aeronet.gsfc-nasa.gov/new_web/data.html. GFED4 fire data is available at https://www.geo.vu.nl/~gwerf/GFED/GFED4/. GPM, TROPOMI, and NOAA/NCEI data can be accessed at https://gpm.nasa.gov/data/directory/, http://www.tropomi.eu/data-products/, and https://www.ncdc.noaa.gov/dataaccess/quick-links. The CAM6-chem simulation data generated in this study have been deposited in the Zenodo repository under open access (https://zenodo.org/record/6050986).

## Code availability

The CESM2 source code can be downloaded from the CESM official website: http://www2.cesm.ucar.edu.

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

## Acknowledgements

W.W. and C.W. are supported by the National Natural Science Foundation of China Grants 41830966. Y.W. is supported by the National Natural Science Foundation of China Grants 41975126 and the National Key Research and Development Program of China Grants 2017YFA0604000. X.L., Z.L., and X.Z. are supported by the US Department of Energy (DOE), Office of Science, Biological and Environmental Research Program (BER), Earth System Modeling and Development Program. We thank Benjamin Gaubert at NCAR for providing BC emissions data used in Supplementary refs. 1, 2. We thank Mingxuan Wu at PNNL for providing MODIS, MISR, and CALIOP DOD data used in Supplementary refs. 3–5 for model evaluation. We thank Louisa Emmons at NCAR for providing gridded SSP245 and COVID-19 emissions data[25] and Adam Phillips and Cecile Hannay at NCAR for providing observed sea surface temperature and sea ice data for the year 2020.

## Author contributions

X.L. and Y.W. conceived and designed the research. L.W. and Y.W. performed the data analysis and data interpretation. X.L. and Z.L. designed the model

experiments, W.W. conducted the model simulations, and X.L., Z.L., W.W., C.W., X.Z., and S.R. performed the model data interpretation. Y.J. and W.X. provided comments and suggestions for the analysis. X.L., Y.W., L.W., and Z.L. wrote the paper. All authors participated in the revision and editing of the paper.

## Competing interests

The authors declare no competing interests.
