## [Peer Review File · Nature Communications]

Black Carbon-Climate Interactions Regulate Dust Burdens over India Revealed during COVID-19Reviewers' Comments:

Reviewer #1:

Remarks to the Author:

This paper is focused on the impacts of the economic lock-down during COVID-19 on changes in burden of absorbing aerosols, i.e., BC and dust, and consequent changes in the climatic state of Indian summer monsoon. Using observation data from multiple sources, including satellite, ground-base platforms, and NASA MERRA2 reanalysis, the authors found large reduction in burden of BC and dust relative to historical records, in conjunction with anomalous cooling and subsidence over northern India, and the formation of an anomalous low-level anticyclone, over the Indian subcontinent. They then conducted numerical experiments using NCAR Community Earth System Model Version 6, with chemistry (CAM6-chem), under various forcing scenarios of aerosol emissions, and concluded that the observed changes in climate states are indeed due to the effects of reduction of BC and dust during COVID-19. Overall, the paper is well-written, and the proposed scenario of changes in climatic states of the Indian monsoon during April 2020 is original and quite plausible. However, there are several areas that need clarification, and additional work. I recommend publication with revisions.

1. It is not clear why the study is only focused in April. The economic lock down in India lasted longer than just April 2020. More important, the Asian monsoon is well known to have large intraseasonal variability. Depending on the pre-monsoon climate states, the evolution of the subsequent ISM system is likely to be different. The authors need to determine if the seasonally colder air in April (relative to JJA) is essential for their proposed mechanism? They should extend their analyses to include April-May-June, or better still the whole monsoon season (MJJAS) to see if the subsequent evolution of the ISM is affected by the April cooling. If they find their proposed mechanism does not work for other months other than April, they should say so.

2. From Fig.1a, dust optical depth (DOD) is already decreasing during the last 4 years (2017-2020). The reduction in DOD in 2020 is not spectacular compared to other years.

Only PM10 shows a significant drop in 2020. Please comments on quality of the data source for PM10. It is still possible that the dust reduction in 2020 April is part of natural variability.

3. Most of the dust loading over northern India during the pre-monsoon period comes from remote transport by the emerging low-level westerlies from the Middle East across the Arabian Sea. The authors focused mostly on local dust source from the Thaar Desert in NW India. Please estimate how much of the change in dust burden is due to reduction of dust emission over the Thaar desert, vs. reduction in remote transport from the Middle East.

4. Given that the focus of this study is in the pre-monsoon month of April, snowdarkening effect (SDE) by changed BC and dust deposition on the Himalayan foothills and the Tibetan Plateau is likely very important. SDE is not emphasized in this study, but should at least be pointed out as an important topic for further studies. Please refer to the vast current literature on snowdarkening effects in affecting the Asian summer monsoon. See examples of recent papers (and references therein) shown below.

Lau and Kim (2018) Impacts of snow-darkening by deposition of light-absorbing aerosols on

snowcover over the Himalayas-Tibetan-Plateau, and influences on the Asian summer monsoon: A possible mechanism for the Blanford Hypothesis. *Atmosphere*, 9, 438, doi:10.3390/atmos9110438

Shi, et al (2019) Snow-darkening versus direct radiative effects of mineral dust aerosol on the Indian summer monsoon onset: role of temperature change over dust sources, *Atmos. Chem. Phys.*, 19, 1605–1622, <https://doi.org/10.5194/acp-19-1605-2019>, 2019.

Jin Q et al (2021) Interaction of mineral dust with the Indian monsoon: Recent advances and challenges. *Earth Science Review*, <https://doi.org/10.1016/j.earscirev.2021.103562>

5. Given the above comments, the schematic in Fig. 4 could be misleading. Clear labeling needs to including in the caption, and in the discussion of the figure that the mechanism is only based on one case study during COVID-19 in April 2020. Possible consequential influences of such pre-conditioning on the evolution of the Indian monsoon are still unknown.

Reviewer #2:

Remarks to the Author:

This paper is novel to state the mechanism of BC-induced decrease in dust in the India due to COVID-19 pandemic. Authors overview lots of satellite observation and reanalysis data to show the evidence of aerosol-radiation-cloud interactions during COVID-19. A series of simulations were conducted to prove the mechanism. The paper was organized and written clearly. The conclusion is some valuable, but it could be more impactful if they can add some discuss about the implication to other desert area or air pollution control by aerosol radiation effect. I suggest adding more quantitative analysis of observation in addition to qualitative description, so that let us know how large the change was observed. The model description is too simple to believe their model result. I suggest adding the detail of characteristics and emission scheme of BC and dust at least. More specific comments are list in the following.

1. Line 60: Although the decrease of dust from 2017 is obvious in Figure 1a, can you show some quantitative evaluation? The DOD was decreasing from 2017 but anthropogenic emission was only taken in 2020. How can you attribute the decrease in 2020 to the emission control due to COVID-19? If so, what's the reason to decrease from 2017 to 2019? What's more interesting for me, the DOD decreased much more sharply in 2011 compared to 2020. Else, based on Figure 1a, DOD in several years before is as low as 2020 but without any anthropogenic emission control. Can you explain the difference and reason to the decrease or low DOD in other historical years?

2. Line 80: the correlation coefficient does not increase significant even though three "abnormal" years are excluded. Do you have any references to explain the "abnormal" years of dust in Indian? Or you need to explain more maybe in the supplementary if you want to exclude them. in How did you calculate the correlation coefficient? By annual average or monthly average or else? Is the correlation statistically significant? Was BC burden changing largely during last 20 years? It seems that most points in Figure 1b gather in the area of -1 to 0. I wonder if the correlation between BC and DU is still positive if only the points in the area of -1 and 0 taken into calculation.

3. Line 83: How about the effect of climate internal variabilities on the concentration of dust in India in April 2020? Can you apportion the contribution from climate variability and anthropogenic emission control to the concentration of dust?

4. Line 85: What did you define "strong events" here?

5. Line 82: Did you exclude the influence of urban dust in DOD ? Would the record low caused by the decrease in urban dust due to the decrease in transportation and human activities during COVID-19 pandemic?

6. Line 111: The increase in the surface albedo would cool the surface, but the reduction of BC in the atmosphere would increase the radiation to the surface although decrease the heating in the low troposphere. So these two BC effects should have opposite effect on the surface temperature. I wonder if they dominate the large decrease in surface temperature on surface. It is amazing to find a decrease in the surface temperature up to 3K. Did you check any surface temperature observation? Did the on-site observation find a large decrease in surface temperature in April 2020 compared to multiyear average?

7. Can you determine the contribution from local and non-local desert to the decrease in the dust concentration in April 2020?

8. Based on Figure S7, modeled DOD is much lower than observations (except for MISR). If the mechanism of dust emission is not good in the model, the conclusion about dust is not reasonable. Did you try other dust emission schemes in the model?

9. Line 153: Although you show lots of figures for the comparison between model and observation (Figure S6-S13), I would suggesting adding a paragraph or section to state the main conclusion of the model evaluation maybe in the supplementary.

10. Line 162: the modeled decrease of surface temperature (0.6K) is much lower than the observed (up to 3K). I wonder the decrease in the surface temperature were dominated by other factors which were not considered in your experiment.
11. Line 169: What does the left contribution of ~15% come from? I'm interested in the comparison between CONTROL and BC_COVID. What is the contribution from other aerosols? Did you consider the emission reduced besides of India which would have effect through long transport?
12. Line 336: how did you set the small perturbation in 12 ensemble simulations? Could you show the detail?
13. Line 338: So is the comparison shown in Figure S7 the result after tuning up or not? What is comparison in April 2020?
14. Is your conclusion of the concurrent reduction of dust and BC also correct in other desert area over the world? Do you find the decrease in the dust in other area? Or any difference between India and other area? Your conclusion will be more valuable if it is correct over the world and will implicate the change of air quality over the world in the future when anthropogenic emission are reduced.
15. What is the optical property of BC set in your model? Did you consider the change in the optical property of aged BC?
16. How much was BC emission reduced in April 2020? How about other aerosols in India?
17. What is the dust emission scheme used in your model? What is the relationship between dust emission and surface wind in your scheme? Is there any other factor affecting the dust emission? Please clarify in the method.
18. Line 349: what is the assumption of dust size distribution and deposition in the model? Details are needed.
19. Line 352: I don't think the underestimation of simulated DOD has small influences on your conclusion. If the underestimation was caused by dust emission scheme or simulated meteorology, the conclusion and mechanism could be changed.

Reviewer #3:

Remarks to the Author:

The authors have considered that Dust is more prevalent in the month of April and impact the northern parts of India. Further, the authors have considered reduction up to 40% in BC due to lockdown which is totally wrong assumption based on Foster et al. 2020 paper published in Nature Climate Change. They have mention decline in BC globally not in India. To the best of my knowledge estimation of BC has not been done in India. I do agree that there was reduction in NO₂. It does not mean reduction in BC. Of course, PM_{2.5} decline during last week of March and April due to lockdown in India and afterwards stated increasing due to crop residue burning in the western parts of India. The main sources of BC in the northern parts of India are power plants and brick kilns and the main source of dust is from Arabia peninsula, and these dusts are frequent and of higher intensity in the month of May and June. I would like to bring to notice of the authors that during 2002 - 2020, the maximum dust events over India are observed in the month of May and June. Due to reduction in AOD and PM_{2.5} from far distance, Himalaya was clearly seen.

Lines 26-27

The strong and precipitous reduction of anthropogenic emissions provides a unique opportunity to disentangle complicated interactions between aerosols and climate.

This was only for short period, such short decline will not much effect unless decline continues for long periods.

The authors state "anticyclonic" which is not convincing. Why have authors considered dust in April? There must be number of papers on dust over India showing long range transport of dust, the Thar Desert area over the years has reduced.

The authors have used some of the published results using ground, AERONET and satellite data and considered decline in BC and climate to bring out record low dust in the month of April which is not true.

There are many recent papers related to Dust, air quality and atmospheric pollution, the authors have

only cited few recent papers, other papers are very relevant when authors have considered BC and DU over India.

Reply to the comments by Reviewer #1

We thank the reviewer for his/her helpful comments and suggestions on improving our manuscript. These comments are incorporated into the manuscript now. Below is our point-by-point response to these comments. The reviewer's comments are in italic and our responses are in normal font.

This paper is focused on the impacts of the economic lock-down during COVID-19 on changes in burden of absorbing aerosols, i.e., BC and dust, and consequent changes in the climatic state of Indian summer monsoon. Using observation data from multiple sources, including satellite, ground-base platforms, and NASA MERRA2 reanalysis, the authors found large reduction in burden of BC and dust relative to historical records, in conjunction with anomalous cooling and subsidence over northern India, and the formation of an anomalous low-level anticyclone, over the Indian subcontinent. They then conducted numerical experiments using NCAR Community Earth System Model Version 6, with chemistry (CAM6-chem), under various forcing scenarios of aerosol emissions, and concluded that the observed changes in climate states are indeed due to the effects of reduction of BC and dust during COVID-19. Overall, the paper is well-written, and the proposed scenario of changes in climatic states of the Indian monsoon during April 2020 is original and quite plausible. However, there are several areas that need clarification, and additional work. I recommend publication with revisions.

Reply: We thank the reviewer for the positive remarks on our work and for the suggestions for further improving the manuscript.

1. It is not clear why the study is only focused in April. The economic lock down in India lasted longer than just April 2020. More important, the Asian monsoon is well known to have large intraseasonal variability. Depending on the pre-monsoon climate states, the evolution of the subsequent ISM system is likely to be different. The authors need to determine if the seasonally colder air in April (relative to JJA) is essential for their proposed mechanism? They should extend their analyses to include April-May-June, or better still the whole monsoon season (MJJAS) to see if the subsequent evolution of the ISM is affected by the April cooling. If they find their proposed mechanism does not work for other months other than April, they should say so.

Reply: Thanks for your valuable suggestions. Following the reviewer's comment, we have extended our analysis to include April-May-June-July-August. We focus on the whole premonsoon season (April-May) of South Asia while the impacts on the subsequent Indian summer monsoon (JJA) are explored as well in the revised paper. The originally proposed mechanisms for April still work for the premonsoon season, which is further updated to include the effects of the anomalous southern Indian BC in addition to those of the anomalous northern Indian BC. By analyzing the correlation coefficient between anomalies (relative to monthly means) of dust (DU) over northern India (averaged over 25°-35° N, 70°-88° E) and monthly BC anomalies at each grid cell

over the domain covering the Indian subcontinent in April-May (Fig. R1.1), we find a dipole pattern with northern Indian BC positively correlated to Indian DU and southern Indian BC negatively correlated to it. The correlations peak over northwestern and southwestern India and can reach up to ± 0.6 . Using the multiple linear regression, DU in northern India is well correlated ($r = 0.75$) with the sum of the northern Indian BC and southern Indian BC. The roles of northern and southern Indian BC in regulating the northern Indian DU are comparable (see regression coefficients in Eq. 1 in the revised paper). We note that neither northern Indian BC nor southern Indian BC alone can better explain DU in northern India ($r=0.5$), implying that northern and southern Indian BC work together to facilitate the Indian DU changes.

Figure R1.1. Correlation coefficients between DU (averaged over northern India as indicated by the square) and BC over each grid cell from MERRA-2. Areas exceeding 90% confidence level are hatched using Student’s t-test.

We also find that the strong cooling in premonsoon season due to the negative BC and DU anomalies in northern India causes a delay in the South Asian summer monsoon outbreak in 2020 (Fig. R1.2c). The Indian summer monsoon precipitation decreases over northern India (Fig. R1.2b and c), which results from the reduced flow of moisture associated with the weakened large-scale advection (Fig. R1.3). These consequential impacts of absorbing aerosol anomalies in the premonsoon season on Indian summer monsoon precipitation are broadly consistent with Lau et al. (2006), but in an opposite direction.

Fig. R1.1 has been included in the revised paper as Fig. 1b. Figs. R1.2-1.3 have been included in the revised SI as Figs. S17-18. The corresponding modifications to the main text can be found in Lines 84-103 and 248-252.

References:

Lau, K. M. & Kim, K. M. Observational relationships between aerosol and Asian monsoon rainfall, and circulation. *Geophysical Research Letters* **33**, (2006).

Figure R1.2. Differences of GPM precipitation for (a) April-May and (b) June-July between 2020 and 2015-2019 mean. (c) Time series of daily GPM precipitation anomalies (relative to the climatological mean of May) over northern India (70°E~88°E, 25°E~35°E as denoted by the square in B) in 2020 (blue) and 2015-2019 (black). Onsets of Indian monsoon outbreak are denoted by symbols (please see revised Methods for the definition of Indian summer monsoon outbreak).

Figure R1.3. June-July differences of column integrated (a) moisture convergence and (b) moisture advection of MERRA-2 between 2020 and 2015-2019.

2. From Fig. 1a, dust optical depth (DOD) is already decreasing during the last 4 years (2017-2020). The reduction in DOD in 2020 is not spectacular compared to other years. Only PM₁₀ shows a significant drop in 2020. Please comments on quality of the data source for PM₁₀. It is still possible that the dust reduction in 2020 April is part of natural variability.

Reply: Following the reviewer's comment, due to the uncertainties with quality of the data source for PM₁₀ (and the short observation period), we removed the observational PM₁₀ in the revised paper. We do not think that the decreasing of DOD is part of natural variability, as DOD variability is well correlated ($r = 0.75$) and regulated by the BC anomalies in India. To demonstrate dust during the premonsoon season in 2020 is not mainly a result of the natural variability, DOD during the premonsoon season in 2021 is included. It begins climbing compared to that in 2020 as seen from MERRA-2 (see updated Fig. 1a in the revised paper) because the reduction of the northern Indian BC is largely eliminated without the lockdown (see updated Fig. S4 in the revised SI). The discussion on this has been added in Lines 84-103 and 123-128 in the revision.

3. Most of the dust loading over northern India during the pre-monsoon period comes from remote transport by the emerging low-level westerlies from the Middle East across the Arabian Sea. The authors focused mostly on local dust source from the Thaar Desert in NW India. Please estimate how much of the change in dust burden is due to reduction of dust emission over the Thaar desert, vs. reduction in remote transport from the Middle East.

Reply: Thanks for your suggestion. Sorry for the misunderstanding as we were not intended to focus mostly on local dust source from the Thar Desert and de-emphasize the remote transport of dust from the Middle East. Following the reviewer's comment, we examined the dust budget in northern India and quantified the contributions to the dust burden change from the remote dust transport from the Middle East versus the local dust emission over the Thar Desert based on MERRA-2 and model simulations. As seen in Figs. 3&S2 in the revision, the changes in the transport terms are larger than the reduction of local dust emission for the change of dust burden. We discuss it in Lines 113-117, 168-171 and 210-213 in the revised paper.

4. *Given that the focus of this study is in the pre-monsoon month of April, snowdarkening effect (SDE) by changed BC and dust deposition on the Himalayan foothills and the Tibetan Plateau is likely very important. SDE is not emphasized in this study, but should at least be pointed out as an important topic for further studies. Please refer to the vast current literature on snowdarkening effects in affecting the Asian summer monsoon. See examples of recent papers (and references therein) shown below.*

Lau and Kim (2018) Impacts of snow-darkening by deposition of light-absorbing aerosols on snowcover over the Himalayas-Tibetan-Plateau, and influences on the Asian summer monsoon: A possible mechanism for the Blanford Hypothesis. Atmosphere, 9, 438, doi:10.3390/atmos9110438

Shi, et al (2019) Snow-darkening versus direct radiative effects of mineral dust aerosol on the Indian summer monsoon onset: role of temperature change over dust sources, Atmos. Chem. Phys., 19, 1605–1622, <https://doi.org/10.5194/acp-19-1605-2019>, 2019.

Jin Q et al (2021) Interaction of mineral dust with the Indian monsoon: Recent advances and challenges. Earth Science Review, <https://doi.org/10.1016/j.earscirev.2021.103562>

Reply: Thanks for bringing our attention to these papers. They are now cited in the revised paper. The importance of snow darkening effect (SDE) is thoroughly highlighted in the introduction and discussion sections. For example, we added the following statement in the introduction: “Accumulation of DU and BC in northern India (Indo–Gangetic Plain, IGP), acting as a heat pump, can reinforce the rainfall during the premonsoon and summer monsoon seasons (Lau and Kim, 2006; Lau et al., 2006). Also, BC and DU deposition on snow over the Tibetan Plateau (TP) can accelerate snow melting and decrease surface albedo (Lau and Kim, 2018; Shi et al., 2019; Jin et al., 2021).”

In the discussion section, we added: “BC reduction over northern India due to the COVID-19 pandemic decreases the solar heating in the atmospheric column and increases the surface albedo of the TP due to the reduced BC snow-darkening effect.”

and “The long-term snow cover over the southern slope of the TP provides the pre-condition for the BC snow darkening effect.”

In addition, we explicitly state that the SDE is considered in the model simulations: “The snow darkening effect due to BC and DU deposition on snow is considered by the Snow, Ice, and Aerosol Radiative (SNICAR) model (Flanner et al., 2007).”

References:

- Lau, K. M. & Kim, K. M. Observational relationships between aerosol and Asian monsoon rainfall, and circulation. *Geophysical Research Letters* **33**, (2006).
- Lau, K. M., Kim, M. K. & Kim, K. M. Asian summer monsoon anomalies induced by aerosol direct forcing: the role of the Tibetan Plateau. *Climate Dynamics* **26**, 855-864 (2006).
- Lau, W. K. M. & Kim, K.-M. Impact of Snow Darkening by Deposition of Light-Absorbing Aerosols on Snow Cover in the Himalayas–Tibetan Plateau and Influence on the Asian Summer Monsoon: A Possible Mechanism for the Blanford Hypothesis. *Atmosphere* **9**, (2018).
- Shi, Z. *et al.* Snow-darkening versus direct radiative effects of mineral dust aerosol on the Indian summer monsoon onset: role of temperature change over dust sources. *Atmos. Chem. Phys.* **19**, 1605-1622 (2019).
- Jin, Q., Wei, J., Lau, W. K. M., Pu, B. & Wang, C. Interactions of Asian mineral dust with Indian summer monsoon: Recent advances and challenges. *Earth-Science Reviews* **215**, 103562 (2021).
- Flanner, M. G., Zender, C. S., Randerson, J. T. & Rasch, P. J. Present-day climate forcing and response from black carbon in snow. *Journal of Geophysical Research: Atmospheres* **112**, (2007).

5. Given the above comments, the schematic in Fig. 4 could be misleading. Clear labeling needs to including in the caption, and in the discussion of the figure that the mechanism is only based on one case study during COVID-19 in April 2020. Possible consequential influences of such pre-conditioning on the evolution of the Indian monsoon are still unknown.

Reply: The caption in the schematic plot is revised to state explicitly that the proposed mechanism works not only for the COVID-19 case, but also is valid for long-term variabilities of dust in northern India based on our analysis of long-term dust statistics. See our reply to your comment #1, we have added studies of the consequential influences of atmospheric cooling dust to BC and dust anomalies in the pre-monsoon season on the Indian summer monsoon.

Reply to the comments by Reviewer #2

We thank the reviewer for his/her comments and suggestions on improving the manuscript. These comments are incorporated into the manuscript now. Below is our point-by-point response. The reviewer's comments are in italic and our responses are in normal font.

This paper is novel to state the mechanism of BC-induced decrease in dust in the India due to COVID-19 pandemic. Authors overview lots of satellite observation and reanalysis data to show the evidence of aerosol-radiation-cloud interactions during COVID-19. A series of simulations were conducted to prove the mechanism. The paper was organized and written clearly. The conclusion is some valuable, but it could be more impactful if they can add some discuss about the implication to other desert area or air pollution control by aerosol radiation effect. I suggest adding more quantitative analysis of observation in addition to qualitative description, so that let us know how large the change was observed. The model description is too simple to believe their model result. I suggest adding the detail of characteristics and emission scheme of BC and dust at least. More specific comments are list in the following.

Reply: We thank the reviewer for pointing out the novelty of our study and for the suggestions for further improving the manuscript. Following the review's comment, we have added more quantitative analysis of observed changes in dust burden. We also added more details on the model description, particularly on the emissions of BC and dust used in the model. The implications of this study to other desert regions or air pollution control by aerosol-radiation interactions are discussed as well.

1. Line 60: Although the decrease of dust from 2017 is obvious in Figure 1a, can you show some quantitative evaluation? The DOD was decreasing from 2017 but anthropogenic emission was only taken in 2020. How can you attribute the decrease in 2020 to the emission control due to COVID-19? If so, what's the reason to decrease from 2017 to 2019? What's more interesting for me, the DOD decreased much more sharply in 2011 compared to 2020. Else, based on Figure 1a, DOD in several years before is as low as 2020 but without any anthropogenic emission control. Can you explain the difference and reason to the decrease or low DOD in other historical years?

Reply: Following the suggestion by reviewer #1, we have extended our analysis on April to the whole premonsoon season (April-May) of South Asia. Also, we have updated our analysis by including the effects of the anomalous southern Indian BC in addition to those of the anomalous northern Indian BC, which significantly improves the correlation of DOD variability with the BC anomalies. By analyzing the correlation coefficient between anomalies (relative to monthly means) of DU over northern India (averaged over 25°-35° N, 70°-88° E) and monthly BC anomalies at each grid cell over the domain covering the Indian subcontinent in April-May (Fig. R2.1), we find a dipole

pattern with northern Indian BC positively correlated to Indian DU and southern Indian BC negatively correlated to it. The correlations over northwestern and southwestern India and can reach up to ± 0.6 . Using the multiple linear regression, DU in northern India is well correlated ($r = 0.75$) with the sum of the northern Indian BC and southern Indian BC. The roles of northern and southern Indian BC in regulating the northern Indian DU are comparable (see regression coefficients in Eq. 1 in the revised paper). We note that the joint effect of northern Indian BC and southern Indian BC can much better explain the DU in northern India than the individual effect of BC ($r=0.5$), implying that northern and southern Indian BC work together to facilitate Indian DU changes. Therefore, in some years despite no reductions of BC in northern India, the minima of northern Indian DU are still found (e.g., in 2011), which can be attributed to the impact of the positive southern BC anomalies resulting from intensified crop residue burning.

Fig. R2.1 has been included in the revised paper as Fig. 1b. The corresponding modifications to the main text can be found in Lines 84-128.

Figure R2.1. Correlation coefficients between DU (averaged over northern India as indicated by the square) and BC over each grid cell from MERRA-2. Areas exceeding 90% confidence level are hatched using the Student’s t-test.

2. Line 80: the correlation coefficient does not increase significant even though three “abnormal” years are excluded. Do you have any references to explain the “abnormal” years of dust in Indian? Or you need to explain more maybe in the supplementary if you want to exclude them. in How did you calculate the correlation coefficient? By annual average or monthly average or else? Is the correlation statistically significant? Was BC burden changing largely during last 20 years? It seems that most points in Figure 1b gather in the area of -1 to 0. I wonder if the correlation between BC and DU is still positive if only the points in the area of -1 and 0 taken into calculation.

Reply: The original Fig. 1b has been replaced with Fig. R2.1 in the revision. We did not divide all the years into normal and abnormal years anymore in the revision. Instead, using 21-year reanalysis data, we analyze the correlation coefficient between monthly anomalies (relative to monthly means of 21 years) of DU over northern India (averaged over 25°-35° N, 70°-88° E) and monthly BC anomalies at each grid cell over the domain covering the Indian subcontinent for April-May (Fig. R2.1). Areas exceeding 90% confidence level are hatched in the figure using the Student's t-test. With the general circulation anomaly induced by anomalous BC during COVID-19, we quantified the contributions from the remote dust transport from the Middle East and the local dust emission over the Thar desert to the DU anomalies based on both MERRA-2 and model simulations. As seen in Figs. 3&S2 in the revision, the contributions of reductions of the zonal transport and local emission are comparable. In addition, since precipitation in the premonsoon season is scarce, the precipitation impact is negligible (see Fig. S17 in the revised SI).

3. Line 83: How about the effect of climate internal variabilities on the concentration of dust in India in April 2020? Can you apportion the contribution from climate variability and anthropogenic emission control to the concentration of dust?

Reply: In the light of Fig. R2.1, we further decompose the long-term time evolution of averaged DU over northern India into averaged BC over northern and southern India during the premonsoon season (April-May) using the multiple linear regression, and obtain the following equation:

$$\Delta DU_{north} = 0.56 \times \Delta BC_{north} - 0.55 \times \Delta BC_{south} + \varepsilon \quad (R2.1)$$

where ΔDU_{north} is the DU anomaly averaged over northern India, ΔBC_{north} is the BC anomaly averaged over northern India (25°-35° N, 70°-88° E), ΔBC_{south} is the BC anomaly averaged over southern India (15°-25° N, 70°-88° E), and ε is the residual. It shows that ΔDU_{north} is well correlated ($r = 0.75$) with the sum of the ΔBC_{north} and ΔBC_{south} . Given $r = 0.75$, ~60% of the northern Indian DU variation can be explained by the northern and southern BC anomalies and the remaining 40% can be attributed to other factors, most of which are climate variabilities.

4. Line 85: What did you define “strong events” here?

Reply: These have been removed in the revision.

5. Line 82: Did you exclude the influence of urban dust in DOD? Would the record low caused by the decrease in urban dust due to the decrease in transportation and human activities during COVID-19 pandemic?

Reply: Urban dust does contribute to DOD especially over India. However, its fractional contribution to total dust is much lower than natural dust (Ginoux et al., 2012). Moreover, since the northern and southern BC anomalies can explain ~60% of variation of the northern Indian DU in long-term reanalysis data, the impact of urban dust on the

dust variation can be small.

References:

Ginoux, P., Prospero, J. M., Gill, T. E., Hsu, N. C. & Zhao, M. Global-scale attribution of anthropogenic and natural dust sources and their emission rates based on MODIS Deep Blue aerosol products. *Reviews of Geophysics* **50**, doi:<https://doi.org/10.1029/2012RG000388> (2012).

6. Line 111: The increase in the surface albedo would cool the surface, but the reduction of BC in the atmosphere would increase the radiation to the surface although decrease the heating in the low troposphere. So these two BC effects should have opposite effect on the surface temperature. I wonder if they dominate the large decrease in surface temperature on surface. It is amazing to find a decrease in the surface temperature up to 3K. Did you check any surface temperature observation? Did the on-site observation find a large decrease in surface temperature in April 2020 compared to multiyear average?

Reply: Over the Tibetan Plateau (TP), the BC reduction in the atmosphere, as the reviewer said, can increase the solar radiation to the surface, which can accelerate snow melting. However, the reduced BC snow darkening effect due to BC deposition in snow can offset this impact. As seen from the increase of surface albedo (Fig. 2b), the latter dominates the snow cover change thus causing a net cooling at the surface. Colder air over the southern slope of the TP moves southward and facilitates the increase of low clouds (Fig. 2d), which reduces the solar radiation to the surface. This reduction overwhelms the increase of solar radiation to the surface due to the reduction of atmospheric aerosols (Fig. S6). Therefore, surface air temperature over northern India declines, which is predominately contributed from the cloud increase.

The observed surface air temperature change is from the National Oceanic and Atmospheric Administration/National Centers for Environmental Information (NOAA/NCEI) which was also used in Yang et al. (2020) for the investigation of the COVID-19 lockdown impact on surface air temperature in China. NOAA/NCEI observations already combining on-site observations show a decrease of up to 2K (now for April-May in the revised paper) compared to multiyear average.

References:

Yang, Y. *et al.* Fast Climate Responses to Aerosol Emission Reductions During the COVID-19 Pandemic. *Geophysical Research Letters* **47**, e2020GL089788 (2020).

7. Can you determine the contribution from local and non-local desert to the decrease in the dust concentration in April 2020?

Reply: We quantified the contributions from the non-local dust transport from the Middle East and the local dust emission over the Thar desert to the decrease of dust burden based on both MERRA-2 and model simulations. As seen in Figs. 3&S2 in the revision, the reductions of the non-local dust transport and local dust emission are comparable. We discuss it in Lines 113-117, 168-171 and 210-213 in the revised paper.

8. *Based on Figure S7, modeled DOD is much lower than observations (except for MISR). If the mechanism of dust emission is not good in the model, the conclusion about dust is not reasonable. Did you try other dust emission schemes in the model?*

Reply: In our previous version of the manuscript, the default dust emission scheme (Zender et al., 2003) in CESM2 significantly underestimates the dust loading over India. Following the reviewer's suggestion, we implemented and tested another dust emission scheme developed by Kok et al. (2014). The results show that the Kok scheme performs much better compared to the default Zender scheme over India as shown in Figure S9a. Therefore, we adopted the Kok scheme in all our new simulations. Consequently, the dust budget analysis is more accurate.

As discussed in the “Model performance” section, the dust burden or DOD is still slightly underestimated compared to observation. We analyze the possible reasons for the underestimation such as “biased assumption of dust particle size and/or overestimated wet and dry deposition.” However, as mentioned in this section, “since we explored the dust change due to BC reduction rather than the absolute dust burdens, the underestimation of the simulated DOD has small influences on the fidelity of the model performance.”

Reference:

Kok, J. F., Mahowald, N. M., Fratini, G., Gillies, J. A., Ishizuka, M., Leys, J. F., et al., An improved dust emission model: Part 1—Model description and comparison against measurements. *Atmospheric Chemistry and Physics*, **14**(23), 13,023–13,041. <https://doi.org/10.5194/acp-14-13023-2014>, (2014)

Zender, C. S., Bian, H. & Newman, D. Mineral Dust Entrainment and Deposition (DEAD) model: Description and 1990s dust climatology. *Journal of Geophysical Research: Atmospheres* **108**, (2003).

9. *Line 153: Although you show lots of figures for the comparison between model and observation (Figure S6-S13), I would suggesting adding a paragraph or section to state the main conclusion of the model evaluation maybe in the supplementary.*

Reply: The figures are renamed as Figures S8-S15 in the revised manuscript. We have added a designated section of “Model performance” summarizing the main findings on the comparison between model and observation. The objective of this section is to show

that modeled fields, including AOD, BC burden, low clouds, liquid water path, atmospheric circulation, shortwave heating rate, air temperature, surface albedo of the TP, and precipitation, are all comparable with observations and/or MERRA-2. In addition, we thoroughly discussed the comparison between the modeled DOD with DOD retrieved from remote sensing techniques (e.g., satellites and AERONET).

10. Line 162: the modeled decrease of surface temperature (0.6K) is much lower than the observed (up to 3K). I wonder the decrease in the surface temperature were dominated by other factors which were not considered in your experiment.

Reply: In our revised manuscript, we conducted two completely new simulation experiments for reproducing the BC dipole pattern as seen in the MERRA-2 result and examined how the BC dipole pattern affects the dust burden in India. The new results showed that the surface temperature is reduced by 1~1.5 K because of such BC dipole feature. Nevertheless, it is lower than the observed value (up to 3 K). In our revised manuscript, we gave an explanation as following: “This can be contributed by many factors, including weaker surface albedo change over TP (Fig. S16d in comparison with Fig. 2b), other absorbing aerosols like brown carbon from CRB (crop residue burning) not considered in the model, and different model years from the MERRA-2 Reanalysis (see Methods).”

11. Line 169: What does the left contribution of ~15% come from? I'm interested in the comparison between CONTROL and BC_COVID. What is the contribution from other aerosols? Did you consider the emission reduced besides of India which would have effect through long transport?

Reply: The left 15% is very likely due to the non-linear effect among different simulations. However, it is very important to point out that, as mentioned in the previous question, we completely redesigned our simulations. In these newly designed cases, we run climatological simulations and conduct 20 ensemble simulations. Here we do not focus on reproducing the 2020 case but would like to examine whether the BC-DU relationship in India holds valid in a manner of climatology as seen by the model. We would like to elucidate the effect of BC dipole feature on dust burden in India, and thus do not consider the reduction of other anthropogenic aerosols inside and outside India in the new simulations. However, other aerosols such as scattering sulfate and organics are not expected to have the similar effect on the dust burden in India as BC since reduction of these scattering aerosols will not lead to southward advection of cold air from the southern slope of the TP.

12. Line 336: how did you set the small perturbation in 12 ensemble simulations? Could you show the detail?

Reply: To address the reviewer's question, we added the following statement in the manuscript: “we conduct 20-member ensemble simulations for both experiments. The

ensemble members are differentiated by a small perturbation in temperature. For each member, we conduct two-year simulation with the second-year simulation used for model analysis and the first-year simulation used as model spin-up.”

13. Line 338: So is the comparison shown in Figure S7 the result after tuning up or not? What is comparison in April 2020?

Reply: The original Figure S7a shows the result of DOD after tuning up. As suggested by the reviewer, we implement and use another dust emission scheme developed by Kok et al. (2014) without any tuning up. The corresponding new figure in the revised manuscript is Figure S9a.

The comparison between modeled DOD and observed DOD (data only available from AEORNET, MISR, and CALIPSO) for 2020 is shown in Figure R2.2. The agreement between modeled and observed DOD is generally good, particularly with MISR DOD.

Figure R2.2: (a) Modeld dust optical depth (DOD) for April and May of 2020 in comparison against DOD observed by (b) AERONET, (c) MISR, and (d) CALIPSO for the same period.

References:

Kok, J. F., Mahowald, N. M., Fratini, G., Gillies, J. A., Ishizuka, M., Leys, J. F., et al. (2014). An improved dust emission model: Part 1—Model description and comparison against measurements. *Atmospheric Chemistry and Physics*, **14**(23), 13,023–13,041. <https://doi.org/10.5194/acp-14-13023-2014>

14. *Is your conclusion of the concurrent reduction of dust and BC also correct in other desert area over the world? Do you find the decrease in the dust in other area? Or any difference between India and other area? Your conclusion will be more valuable if it is correct over the world and will implicate the change of air quality over the world in the future when anthropogenic emission are reduced.*

Reply: South Asia is unique in that it is subject to the strong pollution from both anthropogenic and biomass burning BC and natural dust. It is also located in a region where high TP serves as a barrier to isolate the South Asia from the influence of other regions in the north. The snow darkening effects of BC in TP and the unique BC emission pattern (i.e., BC emitted from industry and transportation over northern India and BC emitted from crop residue burning over southern India) regulate the DU in South Asia. Other desert regions in the world may not have concurrent strong BC emissions. Thus, dust variations in other desert regions over the world may not be directly linked to local BC emission variations. However, we do find that the dust in East Asia (Gobi and Taklamakan desert) may also be perturbed due to the reduction of anthropogenic pollution in 2020 during the lockdown (results not published).

15. *What is the optical property of BC set in your model? Did you consider the change in the optical property of aged BC?*

Reply: We added the following statement in the manuscript: “We use $1.95 - 0.79i$ for the refractive index of BC in model”. In MAM4, aged BC will be transferred from the primary carbon mode to the accumulation mode where it will be internally mixed with other soluble aerosol species, so that the optical property of internally mixed aerosols with BC component is evolved because of the aging process. However, the refractive index of BC remains unchanged because the version of CESM that we use does not deal with a time-evolving refractive index.

16. *How much was BC emission reduced in April 2020? How about other aerosols in India?*

Reply: According to Doumbia et al. (2021), during the 2020 April and May in India, BC emission is reduced by 21.1%, while SO₂ (precursor of sulfate aerosols) is reduced by 30.9%, and NO_x (precursors of nitrate aerosols) is reduced by 35.4% compared to mean values of 2015-2019 April and May. Particulate organic matter (POM) emission is slightly enhanced by 1.9%. According to Forester et al. (2020), these values are -35.7% for BC; -38.3% for SO₂; -44.5% for NO_x, and -26.8% for POM.

Here we want to emphasize that, since we completely redesign the model experiments, we do not consider other aerosols' contributions but only focus on the BC dipole feature's impact on dust burden in India. The magnitude of the BC dipole feature is very close to MERRA-2 results; therefore we are confident to say that “the difference in the

thermodynamic and aerosol fields between the two cases will be interpreted in a similar manner as our observation/MERRA-2 data analyses.”

References:

Doumbia, T. *et al.* Changes in global air pollutant emissions during the COVID-19 pandemic: a dataset for atmospheric chemistry modeling. *Earth System Science Data Discussions* (2021): 1-26.

Forster, P. M. *et al.* Current and future global climate impacts resulting from COVID-19. *Nature Climate Change* 10, 913-919 (2020).

17. *What is the dust emission scheme used in your model? What is the relationship between dust emission and surface wind in your scheme? Is there any other factor affecting the dust emission? Please clarify in the method.*

Reply: Thank you for the comment. The CESM2 used in our study adopts the dust emission scheme of Kok et al. (2014a). As described in Kok et al. (2014a), the dust emission depends on the surface wind speed, soil moisture, and exposed bare soil fraction. The dust emission is proportional to the n -th power of surface wind strength (defined as the friction velocity) and $n=\alpha+2$ where $\alpha (>0)$ is proportional to the relative erodibility of the surface.

We have clarified in the revised manuscript: “The Kok scheme is developed on the basis that sandblasting is the main mechanism for dust emission. The scheme accounts for two processes missing in most existing parameterizations: a soil’s increased ability to produce dust under saltation bombardment as it becomes more erodible, and the increased scaling of the dust flux with wind speed as soil becomes less erodible. The dust emission flux (F) is then calculated from the friction velocity, threshold friction velocity, atmospheric density, clay content in the soil, and the areal fraction of exposed bare soil. Specifically, F is proportional to the n -th power of surface wind strength (defined as the friction velocity) and $n=\alpha+2$ where $\alpha (>0)$ is proportional to the relative erodibility of the surface. The Kok scheme has shown significant improvements over the Zender scheme used in the default CESM2 (Kok et al., 2014b).”

References:

Kok, J. F., Mahowald, N. M., Fratini, G., Gillies, J. A., Ishizuka, M., Leys, J. F., et al. (2014a). An improved dust emission model: Part 1—Model description and comparison against measurements. *Atmospheric Chemistry and Physics*, **14**(23), 13,023–13,041. <https://doi.org/10.5194/acp-14-13023-2014>

Kok, J. F., Albani, S., Mahowald, N. M., & Ward, D. S. (2014b). An improved dust

emission model: Part 2—Evaluation in the Community Earth System Model, with implications for the use of dust source functions. *Atmospheric Chemistry and Physics*, **14**(23), 13,043–13,061. [https://doi.org/10.5194/acp - 14 - 13043 - 2014](https://doi.org/10.5194/acp-14-13043-2014)

Zender, C. S., H. S. Bian, and D. Newman (2003), Mineral Dust Entrainment and Deposition (DEAD) model: Description and 1990s dust climatology, *J Geophys Res-Atmos*, **108**(D14), 4416, doi:10.1029/2002jd002775.

18. Line 349: *what is the assumption of dust size distribution and deposition in the model? Details are needed.*

Reply: Thank you for the comment. The transport and deposition of dust particles are treated by a four-mode version of Modal Aerosol Module (MAM4) used in CESM2, as described in Liu et al. (2012; 2016) and references therein. In particular, emitted dust size distribution is prescribed according to the dust size distribution of Kok (2011). The dust size distribution is evolving in the atmosphere depending on the dust dry and wet deposition processes. Dust dry deposition is treated with the parameterization developed by Zhang et al. (2001). Dust wet deposition includes both in-cloud and below-cloud scavenging, which is described in Liu et al. (2012).

In the revised manuscript, we have added more details on dust in the revised manuscript: “The emitted dust particles are partitioned into Aitken, accumulation, and coarse modes following the dust size distribution of Kok (2011), which is derived on the assumption that breaking of soil particles during dust emissions is analogous to the fragmentation of brittle materials. The size ranges for the three dust modes in diameter are 0.01-0.1 μm , 0.1-1.0 μm , and 1.0-10 μm , respectively and according to Kok (2011), the mass fractions for the three dust modes are 0.00165%, 1.1%, and 99.9%, respectively. Dust deposition includes both dry and wet deposition, which are treated separately. For dry deposition, CESM2 adopts the deposition of Zhang et al. (2001). For wet deposition, the in-cloud and below-cloud scavenging are treated separately, following the parameterizations described in Liu et al. (2012)”.

References:

Kok, J. F.: A scaling theory for the size distribution of emitted dust aerosols suggests climate models underestimate the size of the global dust cycle, *P. Natl. Acad. Sci. USA*, **108**, 1016–1021, <https://doi.org/10.1073/pnas.1014798108>, 2011.

Liu, X., et al. (2012), Toward a minimal representation of aerosols in climate models: Description and evaluation in the Community Atmosphere Model CAM5, *Geosci. Model Dev.*, **5**(3), 709–739, doi:10.5194/gmd-5-709-2012.

Liu, X., Ma, P.-L., Wang, H., Tilmes, S., Singh, B., Easter, R. C., Ghan, S. J., and Rasch, P. J.: Description and evaluation of a new four-mode version of the Modal Aerosol Module (MAM4) within version 5.3 of the Community Atmosphere Model, *Geosci. Model Dev.*, **9**, 505–522, <https://doi.org/10.5194/gmd-9-505-2016>, 2016.

Zhang, L., Gong, S., Padro, J., & Barrie, L. (2001). A size-segregated particle dry deposition scheme for an atmospheric aerosol module. *Atmospheric Environment*, **35**, 549–560. [https://doi.org/10.1016/S1352-2310\(00\)00326-5](https://doi.org/10.1016/S1352-2310(00)00326-5)

19. Line 352: I don't think the underestimation of simulated DOD has small influences on your conclusion. If the underestimation was caused by dust emission scheme or simulated meteorology, the conclusion and mechanism could be changed.

Reply: The underestimation of simulated DOD was caused by the dust emission scheme. As mentioned in comment #13, the DOD is no longer underestimated for 2020 thanks to the replacement of the Zender dust scheme with the Kok dust scheme. With this new dust scheme, we still get the similar results on the BC-DU relationship in India; therefore our conclusions are valid.

Reply to the comments by Reviewer #3

We thank the reviewer for his/her comments and suggestions on improving the manuscript. These comments are incorporated into the manuscript now. Below is our point-by-point response. The reviewer's comments are in italic and our responses are in normal font.

The authors have considered that Dust is more prevalent in the month of April and impact the northern parts of India. Further, the authors have considered reduction up to 40% in BC due to lockdown which is totally wrong assumption based on Foster et al. 2020 paper published in Nature Climate Change. They have mention decline in BC globally not in India. To the best of my knowledge estimation of BC has not been done in India. I do agree that there was reduction in NO₂. It does not mean reduction in BC. Of course, PM_{2.5} decline during last week of March and April due to lockdown in India and afterwards stated increasing due to crop residue burning in the western parts of India. The main sources of BC in the northern parts of India are power plants and brick kilns and the main source of dust is from Arabia peninsula, and these dusts are frequent and of higher intensity in the month of May and June. I would like to bring to notice of the authors that during 2002 - 2020, the maximum dust events over India are observed in the month of May and June. Due to reduction in AOD and PM_{2.5} from far distance, Himalaya was clearly seen.

Reply: We thank the reviewer for the valuable comments. Our previous analysis focused on the month of April because this was the time when the lockdown in India started to generate large impacts on the emissions of air pollutants, although we don't imply that dust in the month of April is more prevalent than those in the months of May and June. Following your and reviewer #1's comments, we have extended our analysis to include the months of April and May (the whole premonsoon season of Aouth Asia). We don't include the month of June in our BC-dust analysis since enhanced precipitation in June plays an important role in the dust variability via wet scavenging of aerosols. However, the impacts of altered premonsoon conditions on the subsequent Indian summer monsoon (JJA) are explored as well in the revised paper.

We examine the dust budget over northern India in April-May based on the MERRA-2 reanalysis data and model simulations, and find that the main source of dust over northern India is indeed from dust transport from Arabia peninsula and the local emission in the Thar Desert only makes a minor contribution (Fig. S2b), consistent with the reviewer's comment. However, the suppressed local dust emission also accounts for a significant fraction to the total dust reduction in northern India during the lockdown (Fig. S2a).

Regarding the BC emission reduction in India, Forster et al. (2020) did show that anthropogenic BC emissions (e.g., from transportation and industry) were reduced in India during COVID-19 (see Fig. 1b in their paper or Fig. R3.1). Other emission

datasets also demonstrate the anthropogenic BC emission reductions in India during COVID-19 (Granier et al., 2019; Doumbia et al., 2021) (Fig. R3.2). Consistent with emission changes, some on-site observations detected the BC concentration reductions near the surface in India during the COVID-19 lockdown (Panda et al., 2020; Sharma et al., 2020; Ambade et al., 2021; Goel et al., 2021, Figure R3.3). That being said, your comment does prompt us to examine the BC emission changes from other sources. Different from northern India with obvious anthropogenic BC reductions, southern and western parts of India during COVID-19 experienced an increase of BC due to intensified crop residue burning (CRB) (Fig. R3.2a for GFED4 and Fig. 1d for MERRA-2 in the revised paper).

Motivated by your comment about CRB, we update our proposed mechanisms in the revision to further include the impact of CRB in southern India. We analyze the correlation coefficient between anomalies (relative to monthly means) of DU over northern India (averaged over 25°-35° N, 70°-88° E) and monthly BC anomalies at each grid cell over the domain covering the Indian subcontinent in April-May (Fig. R3.4). A dipole pattern is noticed with northern Indian BC positively correlated to Indian DU and southern Indian BC negatively correlated to it. The correlations peak over northwestern and southwestern India and can reach up to ± 0.6 . Using the multiple linear regression, DU in northern India is well correlated ($r = 0.75$) with the combination of the northern Indian and southern Indian BC. The roles of northern and southern Indian BC in regulating the northern Indian DU are comparable (see regression coefficients in Eq. 1 in the revised paper). We note that neither northern Indian BC nor southern Indian BC alone can better explain DU in northern India ($r=0.5$), implying that northern and southern Indian BC work together to facilitate the Indian DU changes.

Fig. R3.4 has been included in the revised paper as Fig. 1b. Fig. R3.2 has been included in the revised SI as Fig. S1. The corresponding modifications to the main text can be found in Lines 62-71, 84-122 and 210-213.

References:

- Panda, S., Mallik, C., Nath, J. *et al.* A study on variation of atmospheric pollutants over Bhubaneswar during imposition of nationwide lockdown in India for the COVID-19 pandemic. *Air Qual Atmos Health* **14**, 97–108 (2021). <https://doi.org/10.1007/s11869-020-00916-5>.
- Sharma, S. et al. Effect of restricted emissions during COVID-19 on air quality in India. *Sci Total Environ* **728**, 138878 (2020).
- Ambade, B., Kurwadkar, S., Sankar, T.K. *et al.* Emission reduction of black carbon and polycyclic aromatic hydrocarbons during COVID-19 pandemic lockdown. *Air Qual Atmos Health* **14**, 1081–1095 (2021). <https://doi.org/10.1007/s11869-021-01004-y>.

Goel, V. *et al.* Variations in Black Carbon concentration and sources during COVID-19 lockdown in Delhi. *Chemosphere* **270**, 129435, doi:https://doi.org/10.1016/j.chemosphere.2020.129435 (2021).

Figure R3.1. A breakdown of the April 2020 average global emission reductions compared to a recent year for the different species (from Fig. 2b in Forster et al. 2020).

Figure R3.2. BC emission changes. (a) BC emission change in April-May between 2020 and 2015-2019 for biomass burning from GFED4. (b-f) BC emission changes in April-May 2020 due to lockdown for anthropogenic sectors (not including biomass burning) of (b) total, (c) energy, (d) industry, (e) transport, and (f) residential from Granier et al. (2019) and Doumbia et al. (2021).

Figure R3.3. A figure adopted from the graphical abstract section of Goel, et al. (2021). The figure clearly shows that BC concentration measured over Delhi, Indian significantly reduced from the pre-lockdown phase to the lockdown phase.

Figure R3.4. Correlation coefficients between DU (averaged over northern India as indicated by the square) and BC over each grid cell from MERRA-2. Areas exceeding 90% confidence level are hatched using Student’s t-test.

Lines 26-27 The strong and precipitous reduction of anthropogenic emissions provides a unique opportunity to disentangle complicated interactions between aerosols and climate. This was only for short period, such short decline will not much effect unless decline continues for long periods. The authors state “anticyclonic” which is not convincing. Why have authors considered dust in April? There must be number of papers on dust over India showing long range transport of dust, the Thar Desert area over the years has reduced.

Reply: Our previous analysis focused on the month of April because this was the time when the lockdown in India started to generate large impacts on the emissions of air pollutants. Our scientific objective is to identify the mechanisms for the interactions between aerosols and climate, as the atmospheric system (e.g., clouds, temperature, circulation) responds rather quickly to the aerosol changes, the so-called fast responses of climate system. The climate responses through changing sea surface temperatures take much longer time, the so-called slow responses of climate system, which is not the focus of this study. That being said, following your and reviewer #1's comments, we have extended our analysis to include the whole premonsoon season (April-May) of South Asia and impacts on the subsequent Indian summer monsoon (JJA) are explored as well in the revised paper.

The Indian average of BC emission reduction lasted to July with the most in April-May compared to the previous five years (Fig. R3.5). The role of the dipole anomalous BC pattern over India in regulating dust burdens during the pre-monsoon season (see reply to your comment #1) is verified by long-term Reanalysis and global climate model simulations. The resulting anomalous easterly surface wind linked to anomalous high sea-level pressure is verified in both MERRA-2 Reanalysis and model simulations.

Following your comment, we examine the dust budget over northern India in April-May based on the MERRA-2 reanalysis data and model simulations, and find that the main source of dust over northern India is indeed from Arabia peninsula and the local emission in the Thar Desert only makes a minor contribution (Fig. S2b), consistent with the reviewer's comment. However, the suppressed local dust emission also accounts for a significant fraction to the total dust reduction in northern India during the lockdown (Fig. 3&S2a).

Figure R3.5. Time series of monthly anomalies of Indian BC emissions in 2020 relative

to the previous five years (2015-2019) (emission data from Forster et al. 2020).

References:

Forster, P. M. *et al.* Current and future global climate impacts resulting from COVID-19. *Nature Climate Change* 10, 913-919 (2020).

The authors have used some of the published results using ground, AERONET and satellite data and considered decline in BC and climate to bring out record low dust in the month of April which is not true.

Reply: We would like to note that the low dust in the month of April-May in 2020 is evident from multiple sources of observations, such as AERONET, CALIPSO, and MERRA-2 reanalysis.

Please see the reply to your comment #1 for the mechanism of BC-climate interactions and impact on the atmospheric circulation and thus on long-range dust transport and local dust emission. Our identified mechanism is broadly consistent with the heat-pump hypothesis proposed in Lau et al. (2006), but in an opposite direction.

Lau, K. M. & Kim, K. M. Observational relationships between aerosol and Asian monsoon rainfall, and circulation. *Geophysical Research Letters* 33, (2006).

There are many recent papers related to Dust, air quality and atmospheric pollution, the authors have only cited few recent papers, other papers are very relevant when authors have considered BC and DU over India.

Reply: Thank you for your valuable suggestions. We have cited more papers on dust, air quality and atmospheric pollution in the revised manuscript.

Panda, S., Mallik, C., Nath, J. *et al.* A study on variation of atmospheric pollutants over Bhubaneswar during imposition of nationwide lockdown in India for the COVID-19 pandemic. *Air Qual Atmos Health* 14, 97–108 (2021). <https://doi.org/10.1007/s11869-020-00916-5>.

Sharma, S. *et al.* Effect of restricted emissions during COVID-19 on air quality in India. *Sci Total Environ* 728, 138878 (2020).

Ambade, B., Kurwadkar, S., Sankar, T.K. *et al.* Emission reduction of black carbon and polycyclic aromatic hydrocarbons during COVID-19 pandemic lockdown. *Air Qual Atmos Health* 14, 1081–1095 (2021). <https://doi.org/10.1007/s11869-021-01004-y>.

Goel, V. *et al.* Variations in Black Carbon concentration and sources during COVID-19 lockdown in Delhi. *Chemosphere* 270, 129435, doi:<https://doi.org/10.1016/j.chemosphere.2020.129435> (2021).

- Lau, K. M. & Kim, K. M. Observational relationships between aerosol and Asian monsoon rainfall, and circulation. *Geophysical Research Letters* **33**, (2006).
- Lau, K. M., Kim, M. K. & Kim, K. M. Asian summer monsoon anomalies induced by aerosol direct forcing: the role of the Tibetan Plateau. *Climate Dynamics* **26**, 855-864 (2006).
- Lau, W. K. M. & Kim, K.-M. Impact of Snow Darkening by Deposition of Light-Absorbing Aerosols on Snow Cover in the Himalayas–Tibetan Plateau and Influence on the Asian Summer Monsoon: A Possible Mechanism for the Blanford Hypothesis. *Atmosphere* **9**, (2018).
- Shi, Z. *et al.* Snow-darkening versus direct radiative effects of mineral dust aerosol on the Indian summer monsoon onset: role of temperature change over dust sources. *Atmos. Chem. Phys.* **19**, 1605-1622 (2019).
- Jin, Q., Wei, J., Lau, W. K. M., Pu, B. & Wang, C. Interactions of Asian mineral dust with Indian summer monsoon: Recent advances and challenges. *Earth-Science Reviews* **215**, 103562 (2021).

Reviewers' Comments:

Reviewer #1:

Remarks to the Author:

The revision and overall responses to reviewers' comments are satisfactory. The authors have acknowledged the possible important effects of changes of dust over India due to changes in transport from remote and local sources in the pre-monsoon season of 2020.

However, I have one minor (but important) comment. As shown in their model results (Fig. 3), the change of dust loading is nearly two orders of magnitude larger than that from BC. It is impossible to tell which has the stronger radiation effect. To confirm the importance of BC reduction due to COVID-19 in their model results, it is more instructive to show the solar cooling rate by BC vs. dust in Fig. 3.

Reviewer #2:

Remarks to the Author:

This manuscript was improved largely after addressed reviewers' comments to previous version. Interactions between aerosol and climate as well as anthropogenic aerosols and natural aerosols are critical topics for a long time. This study focused on a typical case to disentangle the fast response of climate system/earth system to the change of anthropogenic aerosols. Comprehensive methods and data, including reanalysis data, remote sensing, and global model, were applied to make their conclusion solid and believable. The manuscript was written and organized very well, which was started by interesting scientific questions and following with deep analysis. I would suggest accepting this study to publish on Nature Communications, although I have following suggestions to improve this paper further just for consideration.

The reason to the increase in the biomass burning in the south India was explained, but why did biomass burning decrease in the north India, which was more notable. The dipole pattern of BC anomalies is the key to the effect on dust and climate. I would like to understand the reason to form the dipole pattern.

This study was wonderful job to take out a comprehensive analysis on the processes of change in dust by emission of BC, but I wonder which is the dominant reason to determine the decrease in the dust? Changes in the absorption effect of BC? Or increase in the albedo of snow cover? Or change in Easterly wind? I think model study is probably able to give the answer.

I would suggest discussing more about the application of this finding to the cases on other years (Line 247), to make the finding during COVID-19 more valuable.

The delay of the South Asian summer monsoon outbreak and changes in the associated precipitation are probably the result of aerosol radiative effect. More detailed discussion on this would make this find more interesting and important.

I would suggest adding more implication from this study to strength the influence of anthropogenic emission on climate changes as well as the interaction between aerosols and cloud. This special case study is very good evidence to reveal aerosol effect on climate.

Reviewer #3:

Remarks to the Author:

What are the noteworthy results?

I do not find any significant results.

Will the work be of significance to the field and related fields?

No

How does it compare to the established literature? If the work is not original, please provide relevant references.

Work is original but the results are not convincing.

Does the work support the conclusions and claims, or is additional evidence needed?

No

Are there any flaws in the data analysis, interpretation and conclusions? Do these prohibit publication or require revision?

Yes, the data shown by the authors are not convincing.

Is the methodology sound? Does the work meet the expected standards in your field?

No

Is there enough detail provided in the methods for the work to be reproduced?

No.

I have gone through the revised manuscript and with the comments made by two other Reviewers and point wise reply by the authors. I have my own reservations with the results presented in the manuscript since I know the region very well and have good knowledge about the dust and black carbon emissions in the northern parts of India.

In the northern parts of India, sources of Black Carbon (BC) are forest fires, crop residue burning and brick kilns and during premonsoon season (March-July) dust is common.

The authors have shown BC and Dust mainly in the northern parts of India, but the title reflects Dust burdens over the South Asia. Since the authors do not discuss about the sources in other parts of South Asia, I will suggest change in the title of the manuscript.

The two Referees have impressed authors to include citations of their work or work of their colleagues. There are many papers published on dust and BC related to the northern parts of India, I do not find citations of these papers by authors.

It is not very clear to me how the authors have mention following statement in the abstract:

- The first sentence – this is not correct to say that the air pollution has heavy black carbon (BC) and dust loading in the premonsoon season. I can understand dust, but I do not agree with BC during premonsoon season. BC due to crop burning is observed starting from the last week of April and dominates in the month of May. During April-May-June, precipitation is not observed generally, there could be some exception. The authors mentioned that the BC-dust analysis since enhanced precipitation in June plays an important role in the dust variability via wet scavenging of aerosols. I do not think this is right. BC is higher during winter season not during premonsoon season. The authors have cited paper by Goel et al. BC is generally lower during premonsoon season, authors may consider showing BC data for the year 2019 before they comment on reduction in BC due to CRB.
- I do not agree with the second sentence that BC has been identified to impact the Indian premonsoon.
- The third sentence – BC climate interactions regulate Indian DU during the premonsoon season. Yes, it is true but is there any relation with BC and Indian DU?
- Further, the next sentence is totally wrong, BC is positively correlated to northern Indian BC and negatively correlated with to the southern Indian BC. I have gone through the manuscript; I do not find any study performed by the authors related to the southern Indian BC. I think the authors referred parts of Southern Asian countries.
- Line 28 – “BC reduction in Northern India due to lockdown decreases the solar heating in the atmosphere and increase the surface albedo of Tibetan Plateau (TP)”. – How the albedo decreases over TP due to lockdown? The BC reduction did not decline during premonsoon season, the emissions from various sources (brick kilns, coal-based power plants, crop residue burning, traffic during premonsoon, but due to complete lockdown in March April emissions from different sources were reduced) of BC in the northern parts of India did not decline.

- Lines 30-33 – this is not right statement. I did not find any reduction in dust emissions reaching to the northern parts of India either from local or long-range transport of dust from Arabia peninsula.
- Lines 34-35 are not supported from the results by the authors.
- Warming of troposphere was observed by Gautam et al. (2009) in the month of May mainly due to dust storms – authors may consult this paper.
- Lines 45-46 – authors have mention deposition of BC and DU over TP, but they did not mention about Himalayan region where BC and dust deposition are also common.
- Lines 46-48 – detection remains challenging due to complicated interactions and feedback. Authors need to explain.
- Lines 49- 59, The authors mention COVID-19 about China, whereas authors have discussed BC and DU over the northern parts of India,
- Lines 59-62, during PM2.5 has increased during May – June, it was reduced during March – April. The AOD has also enhanced during premonsoon season in 2020 compared to 2019 and other years.
- Lines 59-60, Himalayas were clearly seen from distant in the month of April 2020, the cited reference by the authors show clear visibility of Himalaya on 9 April 2021 when PM2.5 was lower due to lockdown, this fact cannot be taken for the end of April and May 2020 to say decline on BC and its relationship with dust. The crop residue burning (CRB) started in the month of April end and May. One can compare with 2019, I do not find any reduction in CRB.
- Lines 62-66, this observation is not true. The authors may carry out analysis and show that their statements are correct.
- Lines 68-71, Not true.
- Lines 89-90, I do not find dipole pattern, the authors have mentioned a dipole pattern is found with northern Indian BC positively correlated to Indian DU and southern Indian BC negatively correlated. Authors must show that the south India is affected by dust before they say dipole pattern. In southern India, BC concentrations is higher, not dust.

Authors may read following papers:

Gautam, et al., 2013. Satellite observations of desert dust-induced Himalayan snow darkening. *Geophys. Res. Lett.* 40, 988–993.

Enhanced pre-monsoon warming over the Himalayan-Gangetic region from 1979 to 2007, Gautam, et al., *Geophysical Research Letters* 36 (7)

Accumulation of aerosols over the Indo-Gangetic plains and southern slopes of the Himalayas: distribution, properties and radiative effects during the 2009 pre-monsoon season, Gautam et al., *Atmospheric Chemistry and Physics* 11 (24), 12841-12863

Premonsoon aerosol characterization and radiative effects over the Indo-Gangetic Plains: Implications for regional climate warming, Gautam, et al., *Journal of Geophysical Research: Atmospheres* 115 (D17)

Aerosol and rainfall variability over the Indian monsoon region: distributions, trends and coupling Gautam, et al., *Annales Geophysicae* 27 (9), 3691-3703

Two contrasting dust-dominant periods over India observed from MODIS and CALIPSO data, Gautam, et al., *Geophysical Research Letters* 36 (6)

Effects of crop residue burning on aerosol properties, plume characteristics, and long-range transport over northern India, DG Kaskaoutis, S Kumar, D Sharma, RP Singh, SK Kharol, M Sharma, ..., *Journal of Geophysical Research: Atmospheres* 119 (9), 5424-5444

Crop residue burning in northern India: increasing threat to greater India, S Sarkar, RP Singh, A Chauhan, *Journal of Geophysical Research: Atmospheres* 123 (13), 6920-6934

Kaskaoutis, et al., 2012, Influence of anomalous dry conditions on aerosols over India: Transport, distribution and properties, *J. Geophys. Res.*, 117, DOI: 10.1029/2011JD017314 Published: MAY 2, 2012

Bhattacharjee, et al. 2007, Influence of a dust storm on carbon monoxide and water vapor over the Indo-Gangetic Plains, *J. Geophys. Res.- Atm.*, v. 112, D18, Article Number: D18203.

Sarkar, et al. 2019, "Impact of Deadly Dust Storms (May 2018) on Air Quality, Meteorological, and Atmospheric Parameters Over the Northern Parts of India", *GeoHealth*, v. 3, Issue: 3, 67-80, DOI: 10.1029/2018GH000170

Prasad, et al. 2006, Influence of coal based thermal power plants on aerosol optical properties in the Indo-Gangetic basin, *Geophys. Res. Lett.*, v. 33 (5), Article Number: L05805.

Reply to the comments by Reviewer #1

We thank the reviewer for his/her helpful comments and suggestions on improving our manuscript. The comments are incorporated into the revised manuscript. Below is our point-by-point response to these comments. The reviewer's comments are in italics, and our responses are in normal font.

The revision and overall responses to reviewers' comments are satisfactory. The authors have acknowledged the possible important effects of changes of dust over India due to changes in transport from remote and local sources in the pre-monsoon season of 2020.

Reply: We thank the reviewer for the positive remarks on our work and further suggestions for improving the manuscript.

However, I have one minor (but important) comment. As shown in their model results (Fig. 3), the change of dust loading is nearly two orders of magnitude larger than that from BC. It is impossible to tell which has the stronger radiation effect. To confirm the importance of BC reduction due to COVID-19 in their model results, it is more instructive to show the solar cooling rate by BC vs. dust in Fig. 3.

Reply: Thank the reviewer for the comment. To answer the reviewer's question, we calculate the solar heating rate by BC and dust with two sets of diagnostic radiation calls, in which the contributions of dust and BC to radiation are removed, respectively. The results in Figure R1.1 show that the reduced BC in northern India ($>25^{\circ}\text{N}$) significantly cools the atmosphere column from surface to about 600 hPa with a magnitude as strong as -0.06 K day^{-1} . In the southern India we see a small warming effect because BC emission is increased in this region by 20%. The cooling effect associated with the dust reduction has a smaller magnitude at about $-0.02 \sim -0.03 \text{ K day}^{-1}$, particularly above 900 hPa level, despite the reduction is two orders of magnitude larger in dust burden than BC burden. This is because of much stronger mass absorption efficiency of BC aerosols due to its larger imaginary part of refractive index. Following the reviewer's suggestion, we added the following text in in the revised text and also added the two figures in Figure 3.

Lines 225-231:

“Fig. 3e shows that the reduced BC in northern India ($>25^{\circ}\text{N}$) significantly cools the atmosphere column from surface to about 600 hPa with a magnitude as strong as -0.06 K day^{-1} . In the southern India we see a small warming effect because BC emission is increased in this region by 20%. The cooling effect associated with the dust reduction has a smaller magnitude at about $-0.02 \sim -0.03 \text{ K day}^{-1}$ (Fig. 3f), particularly above 900 hPa level, despite the reduction in dust burden is about two orders of magnitude larger than that in the BC burden. This is because of much stronger mass absorption efficiency of BC aerosol.”

Figure R1.1. Changes in the heating rate by (a) BC and (b) dust aerosols between COVID-19 case and SSP case.

Reply to the comments by Reviewer #2

We thank the reviewer for his/her comments and suggestions on improving the manuscript. These comments are incorporated into the revised manuscript. Below is our point-by-point response. The reviewer's comments are in italics, and our responses are in normal font.

This manuscript was improved largely after addressed reviewers' comments to previous version. Interactions between aerosol and climate as well as anthropogenic aerosols and natural aerosols are critical topics for a long time. This study focused on a typical case to disentangle the fast response of climate system/earth system to the change of anthropogenic aerosols. Comprehensive methods and data, including reanalysis data, remote sensing, and global model, were applied to make their conclusion solid and believable. The manuscript was written and organized very well, which was started by interesting scientific questions and following with deep analysis. I would suggest accepting this study to publish on Nature Communications, although I have following suggestions to improve this paper further just for consideration.

Reply: Thank you for recommending the publication of our study and for your further comments and suggestions on improving the manuscript.

The reason to the increase in the biomass burning in the south India was explained, but why did biomass burning decrease in the north India, which was more notable. The dipole pattern of BC anomalies is the key to the effect on dust and climate. I would like to understand the reason to form the dipole pattern.

Reply: Different from small-scale individual agriculture waste burning in southern India, crop residue burning (CRB) in northern India (especially in northwestern India) is large-scale and based on collective farming. Due to the COVID-19 lockdown, migrant laborers who used to work in transplanting rice paddies in northwestern India returned to their home states in southern India. This resulted in a decrease in CRB in northern India but an increase in the biomass burning in southern India. We have explained this in Lines 62-64 in the revision.

The dipole pattern denotes the correlation coefficients between the regionally averaged DU (averaged over 25°-35° N, 70°-88° E) in northern India and the BC anomaly in each grid (Fig. R2.1b) rather than the dipole pattern of BC anomalies. Thus, this does not mean that BC concentrations in northern and southern India change oppositely. With the northern BC increase (decrease) or the southern BC decrease (increase) alone, it still favors the increase (decrease) of DU in northern India. If the BC changes over northern and southern India are opposite, the change in northern Indian DU will become much more prominent. The dipole pattern of the correlation coefficients shows a cutoff along ~25°N, which is consistent with the southern boundary of the westerly dust transportation belt (Fig. R2.1a). This is not occasional but is formed by atmospheric BC regulation through its radiative heating effect. For example, if there are decreases in BC in the south of the transportation belt (i.e., south India), it will cool

the air there and produce an anomalous northward pressure gradient force. To satisfy the geostrophic approximation, especially at levels above the near-surface, this must be balanced by an anomalous southward Coriolis force with the emergence of westerly wind anomalies, which promotes the dust transport from the Middle East and Sahara and local emissions from the Thar Desert. We have made it clearer that this dipole pattern is for the positive/negative correlations of northern Indian DU with BC rather than for the BC anomalies and discussed this mechanism more in Lines 80-89 and 171-175 in the revision.

Figure R2.1. (a) 2015-2019 mean of DU transport flux in April-May. (b) Correlation coefficients between northern Indian DU (averaged over the square) and BC over each grid cell of the Indian subcontinent from MERRA-2 in April-May. Areas exceeding the 90% confidence level are hatched using Student's t test.

This study was wonderful job to take out a comprehensive analysis on the processes of change in dust by emission of BC, but I wonder which is the dominant reason to determine the decrease in the dust? Changes in the absorption effect of BC? Or increase in the albedo of snow cover? Or change in Easterly wind? I think model study is probably able to give the answer.

Reply: Thank the reviewer for the comment. To answer the reviewer's question, we conducted another case study, called NOSDE, in which we used the same configuration as the COVID-19 case but turned off the SNICAR model inside CESM2 so that the snow darkening effect (SDE) is turned off. In other words, in the NOSDE case, the surface albedo does not change as absorbing aerosols depositing on snow surface. Like the other two cases, we also conducted 20 ensemble members and averaged the results of April and May of these 20 ensemble members. Figure R2.2a shows the difference in 10 m wind vectors and speed (contour) between the NOSDE case and the COVID-19 case. We see the difference in 10 m wind is westerly with a magnitude of 0.2 m s^{-1} . Given the results shown in Figure 3c for the total BC effect, this indicates that the SDE of BC accounts for about one tenth of easterly wind anomalies. In Figure R2.2b, we show the relative contribution of SDE of BC to dust burden reduction (calculated as $[\text{COVID-19} - \text{NOSDE}] / [\text{COVID-19} - \text{SSP}]$). The term in numerator represents the dust reduction due to SDE only, while the term in denominator represents the dust reduction due to both SDE and ARI). We found that in the northern India, the SDE roughly accounts for less than 5% of dust reduction. In the central part of India (between 15°N and 20°N), SDE contributes to as high as 20% of dust reduction. Averaged over entire India, SDE contributes to 4% of dust reduction. Therefore, the model results indicate that the ARI is the major factor triggering the dust reduction.

Following the reviewer’s suggestion, we added the following text in the text and the two figures as Figure S17 in SI.

Lines 232-238:

“The comparison between the COVID-19 and NOSDE cases shown in Fig. S17a indicates that the SDE of BC accounts for about one tenth of easterly wind anomalies. As shown in Fig. S17b, the SDE roughly accounts for less than 5% of the DU reduction in the northern part of India. In the central part of India (between 15°N and 20°N), the SDE contributes to as high as 20% of the DU reduction. Averaged over entire India, SDE contributes to 4% of the DU reduction. Therefore, the reduced solar heating in the atmospheric column due to BC reduction is the major factor triggering the DU reduction.”

Figure R2.2. (a) Difference in wind (vector) and wind speed (contour) between COVID-19 and NOSDE cases. (b) Relative contribution (%) of SDE of BC to dust burden reduction, which is calculated as dust burden change of [COVID-19 - NOSDE] / [COVID-19 - SSP].

I would suggest discussing more about the application of this finding to the cases on other years (Line 247), to make the finding during COVID-19 more valuable.

Reply: Thanks for the suggestion. To demonstrate that the findings revealed during COVID-19 can be applied to other years, we further discuss the case for 2018, which shows substantially high dust loadings (Figure R2.3). Overall, the BC anomalies and consequences are opposite to those in the COVID-19 case. In contrast to the COVID-19 epidemic in 2020,

both DU and BC increased in northern India compared with the 2015-2019 means (Fig. R2.4a-c). Higher BC loading strengthened the solar heating rate in the atmosphere at 20-30°N (Fig. R2.4d), and anomalous BC decreased the snow albedo over the TP (Fig. R2.4e). Low clouds were reduced, and solar radiation reaching the surface was increased accordingly (Fig. R2.4h&i). As a result, the near-surface 2 m air temperature increased, especially over western India (Fig. R2.4j). Under these circumstances, the strengthened dust transport flux from the Middle East and Sahara contributes to the increase in Indian DU (Fig. R2.4g). Figure R2.4 is included in the revised SI as Figure S18, and the corresponding text was added in Lines 274-283 in the revised manuscript to emphasize that the findings in this study can be applied to other years.

Figure R2.3. Time series of normalized anomalies of dust optical depth (DOD) after removing the long-term trend (DOD_DT, blue) and after subtracting the long-term mean (DOD_DM, red) over India from CALIPSO (2007-2020) (solid lines) as well as the counterparts (DOD_DT in gray and DOD_DM in purple) from MERRA-2 (2007-2021) (dashed lines).

Figure R2.4. BC-induced anomalies in the premonsoon season of 2018 with high northern Indian DU loading. (a) BC burdens in MERRA-2. (b) DU burdens in MERRA-2. (c) AOD of dust in CALIPSO (contour fill) and of coarse mode in AERONET (colored circles). (d) Zonal mean (70°E–90°E) shortwave heating rate in MERRA-2. (e) Surface albedo in MODIS. (f) Zonal mean (70°E–90°E) air temperature (contour fill) and stream function (vectors) in MERRA-2. (g) DU transport flux by zonal wind in MERRA-2. (h) Low cloud fraction in MERRA-2. (i) Surface shortwave cloud radiative effect in CERES. (j) 2-m air temperature in NOAA/NCEI. All anomalies are relative to 2015-2019 multiyear means.

The delay of the South Asian summer monsoon outbreak and changes in the associated precipitation are probably the result of aerosol radiative effect. More detailed discussion on this would make this find more interesting and important.

Reply: Thanks for the comment. The delay of monsoon outbreaks associated with the decrease in precipitation results from the radiative effects of BC and DU. The reduction in BC and DU cools the atmosphere, triggering anomalous anticyclonic circulation in the lower troposphere,

which restrains the convergence of water vapor from the Indian Ocean. This theoretical framework has been mentioned as the elevated heat pump hypothesis (Lau et al., 2006a&b, 2018; Shi et al., 2019; Jin et al., 2021) but with the opposite pattern in this study. We discussed this more in Lines 288-291 in the revision.

References:

- Lau, K. M. & Kim, K. M. Observational relationships between aerosol and Asian monsoon rainfall, and circulation. *Geophysical Research Letters* **33**, (2006a).
- Lau, K. M., Kim, M. K. & Kim, K. M. Asian summer monsoon anomalies induced by aerosol direct forcing: the role of the Tibetan Plateau. *Climate Dynamics* **26**, 855-864 (2006b).
- Lau, W. K. M. & Kim, K.-M. Impact of Snow Darkening by Deposition of Light-Absorbing Aerosols on Snow Cover in the Himalayas–Tibetan Plateau and Influence on the Asian Summer Monsoon: A Possible Mechanism for the Blanford Hypothesis. *Atmosphere* **9**, (2018).
- Shi, Z. *et al.* Snow-darkening versus direct radiative effects of mineral dust aerosol on the Indian summer monsoon onset: role of temperature change over dust sources. *Atmos. Chem. Phys.* **19**, 1605-1622 (2019).
- Jin, Q., Wei, J., Lau, W. K. M., Pu, B. & Wang, C. Interactions of Asian mineral dust with Indian summer monsoon: Recent advances and challenges. *Earth-Science Reviews* **215**, 103562 (2021).

I would suggest adding more implication from this study to strength the influence of anthropogenic emission on climate changes as well as the interaction between aerosols and cloud. This special case study is very good evidence to reveal aerosol effect on climate.

Reply: Thanks for the valuable suggestion. The findings in this study imply that as anthropogenic aerosol and aerosol precursor emissions in India decline in the future, regional dust loading will decrease through BC-climate interactions. The reductions in BC and DU in India may also contribute to the weakening of the Indian summer monsoon in the future. This indicates co-benefits of BC reduction for air quality, human health, and climate change (Yang et al., 2017; Vandyck et al., 2018; Tong et al., 2021). These implications have been discussed in Lines 292-296 in the revision.

References:

- Yang, Y. *et al.* Dust-wind interactions can intensify aerosol pollution over eastern China. *Nature Communications* **8**, 15333, doi:10.1038/ncomms15333 (2017).
- Vandyck, T. *et al.* Air quality co-benefits for human health and agriculture counterbalance costs to meet Paris Agreement pledges. *Nature Communications* **9**, 4939, doi:10.1038/s41467-018-06885-9 (2018).
- Tong, D. *et al.*, Health co-benefits of climate change mitigation depend on strategic power plant retirements and pollution controls. *Nature Climate Change* **11**, 1077-1083, doi:10.1038/s41558-021-01216-1 (2021).

Reply to the comments by Reviewer #3

We thank the reviewer for his/her comments and suggestions on improving the manuscript. These comments are incorporated into our revised manuscript. Below is our point-by-point response. The reviewer's comments are in italics, our responses are in normal font and the original/revised text in the manuscript is in blue.

I have gone through the revised manuscript and with the comments made by two other Reviewers and point wise reply by the authors. I have my own reservations with the results presented in the manuscript since I know the region very well and have good knowledge about the dust and black carbon emissions in the northern parts of India.

In the northern parts of India, sources of Black Carbon (BC) are forest fires, crop residue burning and brick kilns and during premonsoon season (March-July) dust is common.

The authors have shown BC and Dust mainly in the northern parts of India, but the title reflects Dust burdens over the South Asia. Since the authors do not discuss about the sources in other parts of South Asia, I will suggest change in the title of the manuscript.

Reply: Thanks for the suggestion. Following the reviewer's comments, we have made significant efforts to revise the manuscript, with the main modifications to the text summarized as follows:

- (1) Changed "South Asia" in the title to "India";
- (2) Cited the papers by Gautam et al. to discuss the dust impacts on Indian premonsoon and monsoon;
- (3) Discussed the snow-darkening effect of BC and DU in the Himalayas;
- (4) Provided additional observations in the response to show the reductions of AOD and BC lasting until May 2020 over northern India;
- (5) Showed the reduction of biomass burning (crop residue burning) BC emissions in northern India in April and May 2020 based on the QFED data product to be consistent with the BC emission reduction based on the GFED data product; and
- (6) Included the figures showing the delayed outbreak of the Indian Summer Monsoon in 2020 and the related text in the response.

The two Referees have impressed authors to include citations of their work or work of their colleagues. There are many papers published on dust and BC related to the northern parts of India, I do not find citations of these papers by authors.

Authors may read following papers:

Gautam, et al., 2013. Satellite observations of desert dust-induced Himalayan snow darkening. *Geophys. Res. Lett.* 40, 988–993.

Enhanced pre-monsoon warming over the Himalayan-Gangetic region from 1979 to 2007, Gautam, et al., *Geophysical Research Letters* 36 (7)

Accumulation of aerosols over the Indo-Gangetic plains and southern slopes of the Himalayas: distribution, properties and radiative effects during the 2009 pre-monsoon season, Gautam et al., *Atmospheric Chemistry and Physics* 11 (24), 12841-12863

Premonsoon aerosol characterization and radiative effects over the Indo-Gangetic Plains: Implications for regional climate warming, Gautam, et al., *Journal of Geophysical Research: Atmospheres* 115 (D17)

Aerosol and rainfall variability over the Indian monsoon region: distributions, trends and coupling Gautam, et al., *Annales Geophysicae* 27 (9), 3691-3703

Two contrasting dust-dominant periods over India observed from MODIS and CALIPSO data, Gautam, et al., *Geophysical Research Letters* 36 (6)

Effects of crop residue burning on aerosol properties, plume characteristics, and long-range transport over northern India, DG Kaskaoutis, S Kumar, D Sharma, RP Singh, SK Kharol, M Sharma, ..., *Journal of Geophysical Research: Atmospheres* 119 (9), 5424-5444

Crop residue burning in northern India: increasing threat to greater India, S Sarkar, RP Singh, A Chauhan, *Journal of Geophysical Research: Atmospheres* 123 (13), 6920-6934

Kaskaoutis, et al., 2012, Influence of anomalous dry conditions on aerosols over India: Transport, distribution and properties, *J. Geophys. Res.*, 117, DOI: 10.1029/2011JD017314 Published: MAY 2, 2012

Bhattacharjee, et al. 2007, Influence of a dust storm on carbon monoxide and water vapor over the Indo-Gangetic Plains, *J. Geophys. Res.- Atm.*, v. 112, D18, Article Number: D18203.

Sarkar, et al. 2019, “Impact of Deadly Dust Storms (May 2018) on Air Quality, Meteorological, and Atmospheric Parameters Over the Northern Parts of India”, *GeoHealth*, v. 3, Issue: 3, 67-80, DOI: 10.1029/2018GH000170

Prasad, et al. 2006, Influence of coal based thermal power plants on aerosol optical properties in the Indo-Gangetic basin, *Geophys. Res. Lett.*, v. 33 (5), Article Number: L05805.

Reply: Thank you for bringing our attention to these papers. We have done our best to cite them in Lines 39, 41, 44, 137, 140, 275, and 279, respectively in the revision.

The first sentence – this is not correct to say that the air pollution has heavy black carbon (BC) and dust loading in the premonsoon season. I can understand dust, but I do not agree with BC during premonsoon season. BC due to crop burning is observed starting from the last week of April and dominates in the month of May. During April-May-June, precipitation is not observed generally, there could be some exception. The authors mentioned that the BC-dust analysis

since enhanced precipitation in June plays an important role in the dust variability via wet scavenging of aerosols. I do not think this is right. BC is higher during winter season not during premonsoon season. The authors have cited paper by Goel et al. BC is generally lower during premonsoon season, authors may consider showing BC data for the year 2019 before they comment on reduction in BC due to CRB.

Reply: Thanks for the comments. Following the reviewer’s comment, we show the seasonal variations of BC and DU loadings in India for the years of 2015-2019 (Figure R3.1). As the reviewer indicated, BC loading has minima during the pre-monsoon and monsoon seasons in India. However, compared with those in developed countries such as the United States and Western Europe, Indian BC loading in the pre-monsoon season is still higher. Previous studies suggested that the Indian climate system is sensitive to BC perturbations during April-June (Lau et al., 2006a&b, 2018; Shi et al., 2019; Jin et al., 2021). To avoid confusion, we removed “in the premonsoon season” in the first sentence in the Line 20 in the revision.

Figure R3.1. Seasonal cycle of BC and DU burdens from MERRA-2. 2015-2019 averaged (a) BC burdens and (b) DU burdens.

References:

Lau, K. M. & Kim, K. M. Observational relationships between aerosol and Asian monsoon rainfall, and circulation. *Geophysical Research Letters* **33**, (2006a).

Lau, K. M., Kim, M. K. & Kim, K. M. Asian summer monsoon anomalies induced by aerosol direct forcing: the role of the Tibetan Plateau. *Climate Dynamics* **26**, 855-864 (2006b).

Lau, W. K. M. & Kim, K.-M. Impact of Snow Darkening by Deposition of Light-Absorbing Aerosols on Snow Cover in the Himalayas–Tibetan Plateau and Influence on the Asian Summer Monsoon: A Possible Mechanism for the Blanford Hypothesis. *Atmosphere* **9**, (2018).

Shi, Z. *et al.* Snow-darkening versus direct radiative effects of mineral dust aerosol on the Indian summer monsoon onset: role of temperature change over dust sources. *Atmos. Chem. Phys.* **19**, 1605-1622 (2019).

Jin, Q., Wei, J., Lau, W. K. M., Pu, B. & Wang, C. Interactions of Asian mineral dust with Indian summer monsoon: Recent advances and challenges. *Earth-Science Reviews* **215**, 103562 (2021).

I do not agree with the second sentence that BC has been identified to impact the Indian premonsoon.

Reply: It has been documented that BC in the late spring April-May has significant impacts on the Indian climate in the premonsoon and monsoon seasons through direct radiative effects (Lau *et al.*, 2006; Meehl *et al.*, 2008; Wonsick *et al.*, 2014) and snow-darkening effects (Rahimi *et al.*, 2019 and Lau *et al.*, 2010). To avoid confusion, we changed the sentence to: “BC has been identified to significant impact the Indian climate.”

References:

- Lau, K. M., Kim, M. K. & Kim, K. M., Asian summer monsoon anomalies induced by aerosol direct forcing: the role of the Tibetan Plateau. *Climate Dynamics* **26**, 855-864 (2006).
- Meehl, G. A., Arblaster, J. M. & Collins, W. D., Effects of Black Carbon Aerosols on the Indian Monsoon. *Journal of Climate* **21**, 2869-2882, doi:10.1175/2007JCLI1777.1 (2008).
- Wonsick, M. M., Pinker, R. T. & Ma, Y., Investigation of the "elevated heat pump" hypothesis of the Asian monsoon using satellite observations. *Atmos. Chem. Phys.* **14**, 8749-8761, doi:10.5194/acp-14-8749-2014 (2014).
- Rahimi, S. *et al.*, Quantifying snow darkening and atmospheric radiative effects of black carbon and dust on the South Asian monsoon and hydrological cycle: experiments using variable-resolution CESM. *Atmos. Chem. Phys.* **19**, 12025-12049, doi:10.5194/acp-19-12025-2019 (2019).
- Lau, W. K. M., Kim, M.-K., Kim, K.-M. & Lee, W.-S., Enhanced surface warming and accelerated snow melt in the Himalayas and Tibetan Plateau induced by absorbing aerosols. *Environmental Research Letters* **5**, 025204, doi:10.1088/1748-9326/5/2/025204 (2010).

The third sentence – BC climate interactions regulate Indian DU during the premonsoon season. Yes, it is true but is there any relation with BC and Indian DU?

Reply: The relation between BC and DU in India is actually the key finding of this study. DU is mostly a natural type of aerosol that is highly affected by climate perturbations (e.g., precipitation and winds). The impacts of BC on the local climate in India have been noted. Therefore, we would like to ask whether the BC-induced climate change can impact Indian dust? In this study we demonstrate the role of northern and southern Indian BC in regulating the Indian climate (e.g., atmospheric heating, snow albedo, clouds, near-surface air temperature, and surface wind). With altered large-scale circulation, dust transport from the Middle East and Sahara and local dust emissions over the Thar Desert are changed, thus influencing local DU loading in India. We have made this clearer in Lines 21-22 in the revised manuscript.

We modified the third sentence to: “However, whether BC-climate interactions regulate Indian DU during the premonsoon season is unclear.”

Further, the next sentence is totally wrong, BC is positively correlated to northern Indian BC and negatively correlated with to the southern Indian BC. I have gone through the manuscript; I do not find any study performed by the authors related to the southern Indian BC. I think the authors referred parts of Southern Asian countries.

Reply: There is a typo in the reviewer’s comment. “BC is positively...” should be “dust is positive...”. In this round of the review, we have shown clearly the important role of southern Indian BC in observations/reanalysis (Figures 1b and Equation 1 in the manuscript) and verified this (Figure 3) by performing model experiments through decreasing northern Indian BC and increasing southern Indian BC emissions for the COVID-19 case. The related text discussing the role of southern Indian BC in the second-round version of the manuscript is quoted below:

Lines 82-101:

“To examine whether BC and DU change concurrently in the premonsoon season, a long-term time series of BC and DU burdens from 2000 to 2020 is applied from the Modern-Era Retrospective analysis for Research and Applications version 2 (MERRA-2) Reanalysis. We analyze the correlation coefficient of anomalous time series (relative to monthly means) between DU over northern India (averaged over 25°-35° N, 70°-88° E) and BC at each grid cell over the domain covering the Indian subcontinent in April-May. A dipole pattern of correlation is found with northern Indian BC positively correlated and southern Indian BC negatively correlated to the northern Indian DU. The correlations peak over northwestern and southwestern India and can reach up to ± 0.6 (Fig. 1b). We further decompose the time evolution of averaged DU over northern India into averaged BC over northern and southern India during the premonsoon season using the multiple linear regression, and obtain the following equation:

$$\Delta DU_{north} = 0.56 \times \Delta BC_{north} - 0.55 \times \Delta BC_{south} + \varepsilon \quad (1)$$

where ΔDU_{north} is the DU anomaly averaged over northern India, ΔBC_{north} is the BC anomaly averaged over northern India (25°-35° N, 70°-88° E), ΔBC_{south} is the BC anomaly averaged over southern India (15°-25° N, 70°-88° E), and ε is the residual. It shows that ΔDU_{north} is well correlated ($r = 0.75$) with the sum of the ΔBC_{north} and ΔBC_{south} . Seen from the two regression coefficients, the roles of northern and southern Indian BC in regulating the northern Indian DU are comparable. We note that neither ΔBC_{north} or ΔBC_{south} alone can explain ΔDU_{north} ($r=0.5$) as well as their combination, implying that northern and southern Indian BC work together to facilitate Indian DU changes.”

Lines 106-107:

“Meanwhile, the southern Indian BC burden increases (Fig. 1d) in April-May 2020 due to the intensified CRB (Fig. S1a).”

Lines 119-120:

“However, AOD in southern India is only slightly changed due to the increase in biomass burning emissions there.”

Lines 121-126:

“As a comparison, the BC burden in April-May of 2021 without the COVID-19 lockdown increases over the Indian subcontinent compared to that of 2020 and is even larger than the 2015-2019 climatological mean (Fig. S4a and c). As predicted by Eq. 1, DU over northern India in April-May 2021 begins climbing as seen from MERRA-2 (Fig. 1a and Fig. S4b). Since there are still positive BC anomalies over southern India, the negative anomalies of northern Indian DU relative to the 2015-2019 climatology remain (Fig. S4d).”

Lines 148-150:

“In comparison with northern India, BC increases from intensified CRB in southern India heat the lower atmosphere and decrease the atmospheric stability (Figs. 2a and S5b), which is unfavorable for the low cloud formation (Fig. 2d).”

Lines 154-156:

“Over southern India, net shortwave radiation by aerosols at the surface slightly decreases (Fig. S6d) due to elevated CRB emissions.”

Lines 163-171:

“In comparison with negative northern Indian BC anomalies, the positive southern Indian BC anomalies are equally important for establishing the meridionally asymmetric cooling, which contributes to the easterly wind anomalies. The cooling over northern India resulting from BC reduction and warming over southern India from increased CRB BC emissions induce an anomalous southward pressure gradient force. To satisfy the geostrophic approximation especially at levels above the near-surface, this must be balanced by an anomalous northward Coriolis force as the emergence of easterly wind anomalies. Owing to anomalous easterly wind, the eastward transport of dust from the Middle East and Sahara as well as local dust emissions in the Thar Desert are suppressed (Figs. S7b and S2a), leading to a record low of dust loading in 2020.”

In the revision, we discussed the role of southern Indian BC more in Lines 171-175:

“The distinct roles of northern and southern BC in regulating the northern Indian DU show a cutoff along $\sim 25^\circ\text{N}$, which is consistent with the southern boundary of the westerly dust transport belt (Figs. 1b and S7a). This is formed by the opposite anomalous meridional pressure gradient forces induced by BC radiative heating in the northern and southern India.”

Lines 182-185:

“In the COVID-19 case, to represent the decreases and increases of BC emissions in northern and southern India respectively, we only use the Forster COVID-19 BC emission inventories²⁴ in northern India while using 120% of SSP245 BC emission inventories in southern India (see Methods).”

Line 28 – “BC reduction in Northern India due to lockdown decreases the solar heating in the atmosphere and increase the surface albedo of Tibetan Plateau (TP)”. – How the albedo decreases over TP due to lockdown? The BC reduction did not decline during premonsoon season, the emissions from various sources (brick kilns, coal-based power plants, crop residue burning, traffic during premonsoon, but due to complete lockdown in March April emissions from different sources were reduced) of BC in the northern parts of India did not decline.

Reply: There may be a typo in the reviewer’s comment: “How the albedo *decreases* over TP due to lockdown?” “decreases” here should be “increases” because we find an increase of the surface albedo of TP, as quoted by the reviewer. Surface albedo over TP from MODIS observations is increased due to the decrease of absorbing BC and DU in 2020 and thus less deposition of BC and DU on snow (Fig. 2b, here Fig. R3.2).

There have been many studies noting the BC decline over northern India in April and May 2020 compared to earlier years. In the first round of the response, we provided some of them to support the BC decline in northern India. Here, we show additional studies (Gogoi et al., 2021; Hudda *et al.*, 2020). As shown in Fig. R3.3 (the first three rows for the northern Indian sites), surface observational stations in northern India all feature the decline of BC in April and May 2020 compared to April and May 2015-2019. Additionally, Fig. R3.4 shows that the decrease in BC can last till May 2020. All of these results are consistent with the BC emission changes (Fig. S1, here Fig. R3.5), Granier et al. (2019), and Doumbia et al. (2021). With the reduced snow-darkening effects of BC (Rahimi et al., 2019; Lau et al., 2010), the decrease in BC leads to the increase of surface albedo over the TP.

Figure R3.2. BC induced climate anomalies in the premonsoon season of 2020 during the COVID-19 pandemic. (b) Surface albedo in MODIS.

Figure R3.3. Day-to-day variability in BC mass concentrations during 2020 (for the period from January to May), along with the average values of the years 2015–2019. The shaded color indicates the standard deviations of the mean of the historical values (from Fig. 4 in Gogoi *et al.*, 2021).

Figure R3.4. Tukey box plots of daily measurements (March 24–May 15) of NO₂, PM_{2.5}, and BC for each of six years at two regulatory monitoring sites in Boston. Black boxes represent the lockdown period. All boxes represent the interquartile range from the 25th to 75th percentiles, whiskers represent the 5th and 95th percentiles, and outliers are shown as dots beyond the whiskers (from Fig. 7 in Hudda *et al.*, 2020).

Figure R3.5. BC emission changes. (a) BC emission change in April-May between 2020 and 2015-2019 for biomass burning from GFED4. (b-f) BC emission changes in April-May 2020 due to lockdown for anthropogenic sectors (not including biomass burning) of (b) total, (c) energy, (d) industry, (e) transport, and (f) residential sectors from Granier *et al.*, 2019 and Doumbia *et al.*, 2021.

References:

- Gogoi, M. *et al.* Response of ambient BC concentration across the Indian region to the nation-wide lockdown: results from the ARFINET measurements of ISRO-GBP. *Current Science* **120**, 341-351 (2021).
- Hudda, N., Simon, M. C., Patton, A. P. & Durant, J. L. Reductions in traffic-related black carbon and ultrafine particle number concentrations in an urban neighborhood during the COVID-19 pandemic. *Science of The Total Environment* **742**, 140931, doi:https://doi.org/10.1016/j.scitotenv.2020.140931 (2020).
- Granier, C., Darras, S. *et al.* The Copernicus Atmosphere Monitoring Service global and regional emissions (April 2019 version). *Copernicus Atmosphere Monitoring Service* **ffhal-02322431v2f**, (2019).
- Doumbia, T. *et al.* Changes in global air pollutant emissions during the COVID-19 pandemic: a dataset for atmospheric chemistry modeling. *Earth Syst. Sci. Data Discuss.* **2021**, 1-26 (2021).

Lines 30-33 – this is not right statement. I did not find any reduction in dust emissions reaching to the northern parts of India either from local or long-range transport of dust from Arabia peninsula.

Reply: We have shown in the paper the reductions of local dust emissions and dust long-range transport from the Middle East and Sahara in 2020 over northern India (Figure 1a; Figure S7). We are attaching these figures here for the reviewer’s convenience (Figures R3.6-3.7). Figure R3.6 from the observations of CALIPSO and MERRA-2 clearly shows the reductions of DOD in 2020 (Fig. R3.6a) with the maximum decrease of DU burden over the Thar desert (Fig. R3.6b), which implies local DU emission reductions (our model simulation further confirms this as shown in Fig. 3d in the manuscript). The decline of DU long-range transport from the Middle East and Sahara is also obvious as seen by MERRA-2 (Fig. R3.7b).

Figure R3.6. DU for the premonsoon season. (a) Time series of normalized anomalies of dust optical depth (DOD) after removing the long-term trend (DOD_DT, blue) and after subtracting the long-term mean (DOD_DM, red) over India from CALIPSO (2007-2020) (solid lines) as well as the counterparts (DOD_DT in grey and DOD_DM in purple) from MERRA-2 (2007-2021) (dashed lines). **(b)** DU burdens in MERRA-2 between 2020 and 2015-2019.

Fig R3.7. (a) 2015-2019 mean of DU transport flux in April-May. (b) Changes in DU transport flux by zonal wind in April-May between 2020 and 2015-2019.

Lines 34-35 are not supported from the results by the authors.

Reply: The discussion on the subsequent impacts of BC and DU changes on the monsoon outbreak was requested by Reviewer #1 in the last review. The delay of the South Asian summer monsoon outbreak is shown in Figs. S17 in the supplementary material, which is consistent with Lau and Kim (2006) but in the opposite direction. Again, we attached them in the response for the reviewer’s convenience (Figs. R3.8). The corresponding text is in Lines 284-288 in the manuscript:

“With negative BC and DU anomalies in northern India during the premonsoon season, a delay of the Indian summer monsoon outbreak in 2020 is found (Fig. S19c) (see Methods). As a result, decreased precipitation emerges over northern India in the following months (Fig. S19b and c)¹³, which results from the declined flow of moisture from the Indian Ocean associated with weakened large-scale advection (Fig. S20).”

References:

Lau, K. M. & Kim, K. M. Observational relationships between aerosol and Asian monsoon rainfall, and circulation. *Geophysical Research Letters* **33**, (2006).

Figure R3.8. Differences in GPM precipitation for (a) April-May and (b) June-July between the 2020 and 2015-2019 means. (c) Time series of daily GPM precipitation anomalies (relative to the climatological mean of May) over northern India ($70^{\circ}\text{E}\sim 88^{\circ}\text{E}$, $25^{\circ}\text{N}\sim 35^{\circ}\text{N}$ as denoted by the square in B) in 2020 (blue) and 2015-2019 (black). Onsets of Indian monsoon outbreaks are denoted by symbols (please see revised Methods for the definition of Indian summer monsoon outbreaks).

Warming of troposphere was observed by Gautam et al. (2009) in the month of May mainly due to dust storms – authors may consult this paper.

Reply: Thank you for bringing our attention to this paper. It was cited in Line 140 in the revision.

Lines 45-46 – authors have mention deposition of BC and DU over TP, but they did not mention about Himalayan region where BC and dust deposition are also common.

Reply: Thanks for the comment. We mentioned it in Lines 43-44 in the revision.

“Also, BC and DU deposition on snow over the Tibetan Plateau (TP) and the Himalayan region can accelerate snow melting and decrease surface albedo¹⁵⁻¹⁸.”

Lines 46-48 – detection remains challenging due to complicated interactions and feedback. Authors need to explain.

Reply: We thank the reviewer for the comment. BC and DU aerosols can induce changes in the climate system (e.g., clouds, land/sea ice). The feedbacks of climate changes could amplify or dampen the impacts of BC and DU. Additionally, natural variability (e.g., ENSO) sometimes overwhelms the signals of the climate impacts of BC and DU. We have explained this in Lines 44-46 in the revision.

“However, detecting the BC and DU effects on the Indian climate remains challenging due to climate feedbacks and natural climate variability^{19,20}.”

Lines 49- 59, The authors mention COVID-19 about China, whereas authors have discussed BC and DU over the northern parts of India,

Reply: We thank the reviewer for the comment. We mention COVID-19 in China because there have been many studies focusing on the impacts of aerosol emission reductions in China during the COVID-19 epidemic. In contrast, few studies pay attention to the situation in India. Therefore, this motivates our study.

Lines 59-62, during PM2.5 has increased during May – June, it was reduced during March – April. The AOD has also enhanced during premonsoon season in 2020 compared to 2019 and other years.

Reply: We thank the reviewer for the comment. In this study, we focus on April and May 2020. As shown in Figures 1f&S3, multiple observations all feature that AOD in northern India declines in April-May 2020 compared to the 2015-2019 means. Here, we further look at April and May 2020 separately. In May 2020, the decrease of AOD is still evident from multiple observations (Figure R3.9).

Following the reviewer’s comment, we compare the Indian AOD in April-May 2020 to that in April-May 2019 from multiple observations. We can see that AOD in northern India was decreased for April and May 2020, especially for the AERONET observations (Figure R3.10).

Figure R3.9. Differences in (a&b) aerosol optical depth (AOD at 550 nm) from MISR (contour fill) and AERONET (colored circles), (c&d) AOD from MERRA-2 and (e&f) from MODIS between 2020 and 2015-2019 for April (left) and May (right), respectively.

Figure R3.10. Same as Fig. R3.9 but for differences between 2020 and 2019 (2020 minus 2019).

Lines 59-60, Himalayas were clearly seen from distant in the month of April 2020, the cited

reference by the authors show clear visibility of Himalaya on 9 April 2021 when PM_{2.5} was lower due to lockdown, this fact cannot be taken for the end of April and May 2020 to say decline on BC and its relationship with dust. The crop residue burning (CRB) started in the month of April end and May. One can compare with 2019, I do not find any reduction in CRB.

Reply: We thank the reviewer for the comment. In Fig. S1a of our paper, we have shown that the CRB from GFED in northern India in April-May 2020 declined compared to that in April-May 2015-2019. Here, we further include another biomass burning observation product: Quick Fire Emissions Dataset (QFED) as well as show April and May 2020 separately. Both products (QFED and GFED) feature a decline in April and May 2020 compared to those in April and May 2015-2019 (Fig. R3.11). We further compare with 2019, and find that there were still CRB reductions in northern India (especially in the northwestern part) for April and May 2020, as shown by both QFED and GFED (Fig. R3.12).

Figure R3.11. BC emission change between 2020 and 2015-2019 for biomass burning from (a&b) QFED2.1 and (c&d) GFED4.1 for (a&c) April and (b&d) May, respectively.

Figure R3.12. Same as Fig. R3.11 but for the change between 2020 and 2019 (2020 minus 2019).

Lines 62-66, this observation is not true. The authors may carry out analysis and show that their statements are correct.

Reply: We thank the reviewer for the comment. We have carried out analysis to show that CRB in April-May 2020 over southern India increased while that over northern India decreased compared to earlier years (Figures R3.11 & 3.12). Please see our reply to your last comment.

Lines 68-71, Not true.

Reply: Our statement is based on the analysis of CALIPSO observations. We respectfully ask the reviewer to provide the evidence/justification for his/her assessment why our analysis of dust reduction in April-May 2020 based on the CALIPSO observations is not true.

Lines 89-90, I do not find dipole pattern, the authors have mentioned a dipole pattern is found with northern Indian BC positively correlated to Indian DU and southern Indian BC negatively correlated. Authors must show that the south India is affected by dust before they say dipole pattern. In southern India, BC concentrations is higher, not dust.

Reply: We thank the reviewer for the comment. The dipole pattern denotes the correlation between the northern Indian DU (averaged over 25°-35° N, 70°-88° E) and the BC anomalies in southern and northern India rather than the dipole pattern of the correlation between the DU and BC anomalies in southern and northern India. We are sorry for the confusion. The dipole pattern is shown in Fig. 1b in this round of the review (attached here as Fig. R3.13b). Additionally, as indicated in Eq. (1) in the text, southern Indian BC affects northern Indian DU (not southern Indian DU). The related text in this round of the review is in Lines 82-101:

“To examine whether BC and DU change concurrently in the premonsoon season, a long-term time series of BC and DU burdens from 2000 to 2020 is applied from the Modern-Era Retrospective analysis for Research and Applications version 2 (MERRA-2) Reanalysis. We analyze the correlation coefficient of anomalous time series (relative to monthly means) between DU over northern India (averaged over 25°-35° N, 70°-88° E) and BC at each grid cell over the domain covering the Indian subcontinent in April-May. A dipole pattern of correlation is found with northern Indian BC positively correlated and southern Indian BC negatively correlated to the northern Indian DU. The correlations peak over northwestern and southwestern India and can reach up to ± 0.6 (Fig. 1b). We further decompose the time evolution of averaged DU over northern India into averaged BC over northern and southern India during the premonsoon season using the multiple linear regression, and obtain the following equation:

$$\Delta DU_{north} = 0.56 \times \Delta BC_{north} - 0.55 \times \Delta BC_{south} + \varepsilon \quad (1)$$

where ΔDU_{north} is the DU anomaly averaged over northern India, ΔBC_{north} is the BC anomaly averaged over northern India (25°-35° N, 70°-88° E), ΔBC_{south} is the BC anomaly averaged over southern India (15°-25° N, 70°-88° E), and ε is the residual. It shows that ΔDU_{north} is well correlated ($r = 0.75$) with the sum of the ΔBC_{north} and ΔBC_{south} . As seen from the two regression coefficients, the roles of northern and southern Indian BC in regulating the northern Indian DU are comparable. We note that neither ΔBC_{north} or ΔBC_{south} alone can explain ΔDU_{north} ($r=0.5$) as well as their combination, implying that northern and southern Indian BC work together to facilitate Indian DU changes.”

Figure R3.13. (a) 2015-2019 mean of DU transport flux in April-May. (b) Correlation coefficients between northern Indian DU (averaged over the square) and BC over each grid

cell of the Indian subcontinent from MERRA-2 in April-May. Areas exceeding the 90% confidence level are hatched using Student's t test.

Reviewers' Comments:

Reviewer #2:

Remarks to the Author:

Authors' response has addressed all of my comments and suggestions. I don't have further comment on this version of paper. I believe this study will be very helpful to understand the impact of aerosol on climate change in the Indian, even over the world. The paper in this version has stated the interactions between BC, dust and climate during the premonsoon season clearly and in depth using comprehensive methods including observations and modelling. I think this paper is good to be published on Nature Communications now and will have high impact.

Reply to the comment by Reviewer #2

We thank the reviewer for his/her remarks on our manuscript. Below is our point-by-point response to the comments. The reviewer's comment is in italics, and our response is in normal font.

Authors' response has addressed all of my comments and suggestions. I don't have further comment on this version of paper. I believe this study will be very helpful to understand the impact of aerosol on climate change in the Indian, even over the world. The paper in this version has stated the interactions between BC, dust and climate during the premonsoon season clearly and in depth using comprehensive methods including observations and modelling. I think this paper is good to be published on Nature Communications now and will have high impact.

Reply: We thank the reviewer for the positive comments and for recommending the publication of our paper in *Nature Communications*.